# Bioplastic Floss of a Novel Microwave-Thermospun Shellac: Synthesis and Bleaching for Some Dental Applications

**DOI:** 10.3390/polym15010142

**Published:** 2022-12-28

**Authors:** Sherif S. Hindi, Uthman M. Dawoud, Khalid A. Asiry

**Affiliations:** 1Department of Arid Land Agriculture, Faculty of Environmental Sciences, King Abdullaziz University (KAU), Jeddah 21589, Saudi Arabia; 2Department of Chemical and Materials Engineering, Faculty of Engineering, King Abdullaziz University (KAU), Jeddah 21589, Saudi Arabia

**Keywords:** shellac, thermoplastic resins, thermospinning process, centrifugation process, microwave thermospinner, dental applications

## Abstract

In this paper, crude flakes (CFs) of shellac were converted into purified, nonwoven, thermospun fibers (shellac floss) using two devices, namely, an electric thermospinner (ETS) and a microwave thermospinner (MTS). This conversion was achieved by the action of heating and the centrifugal forces that arose toward the outside of the spinner-head cavity. The dissolved MTS floss was bleached using hydrogen peroxide to produce the bleached MTS floss. The unbleached shellac (CFs, ETS floss, and MTS floss) and the bleached MTS floss were characterized physically and chemically. There was no deterioration in the floss properties due to the heating tools or bleaching process. For the unbleached shellac, although there were no statistical differences in properties among the three shellac types (CFs, ETS floss, and MTS floss), except for insolubility in hot alcohol, acid value, and moisture content, the MTS floss exhibited superior values compared with the other types for nearly all the properties studied. Bleaching the MTS floss produced the greatest color change among other studies, caused a high reduction in insoluble solid matter due to increasing the solubility of some of the solid constituents of shellac, and slightly decreased its Young’s modulus (*E*). The important dental applications were surveyed and it was suggested that the suitability was enhanced by using the bleached MTS floss, based on its superior whiteness, along with the unique properties detected.

## 1. Introduction

Shellac is a natural, polymeric resin secreted by the female lac insect (*Laccifer lacca*), which can be found on certain trees, especially *Schleichera oleosa*, *Schleichera oleos*, *Ziziphus Mauritania*, and *Butea monosperma*. The immature nymphs exude a resin/wax mixture through their bodies after they suck sap from the tree, forming a defensive shell as a cocoon [1]. Shellac is processed, purified, and then sold as dry flakes [2,3,4,5]. It can be produced either as membranes, flakes, buttons, dyes, wax, or refined products, including waxed, dewaxed, and bleached liquids [6,7,8,9]. In addition, fibrous shellac can be synthesized from the crude flakes (CFs) of shellac using ETS and MTS thermospinners [10] and can also be produced by electro-spinning [11,12].

The CFs are composed of resin (approximately 68%), wax (approximately 6%), around 1% dye, and around 25% contaminants [5,13]. Chemically, shellac resin comprises soft resin (ether-soluble monoesters) and hard resin (ether-insoluble complex) at a ratio of approximately 1:3 [14,15]. The orientation of the polar groups of soft resin and hard resin, containing hydroxy acids, is responsible for the strong adherence of shellac to smooth surfaces [3,6]. The main structure of shellac consists of polyesters and single esters, which form hydroxyl and carboxyl groups [16,17].

Microwaves are a part of the broad electromagnetic radiation spectrum, having a dual nature since it is both wave-like and particle-like [18]. A typical microwave apparatus consists mainly of a microwave emitter (magnetron). Their industrial applications include thermal treating, curing polymers, the synthesis of chemical compounds, and reaction acceleration.

Shellac has gained importance in the food and medicinal fields due to its unique non-toxic and hypoallergenic properties. Shellac is biodegradable, film-forming, has excellent adhesion, hardness, and high gloss; it is superior in terms of electrical properties and is compatible with other resins [19]. Furthermore, it has superior moisture protection, so it is useful for enhancing moisture-sensitive products and extending their shelf life [19,20,21]. Shellac is used as a tablet coating for slow-release drugs in pharmaceutical applications [22,23], as an essential binder in cosmetics, in the micro-encapsulation of fragrances, as a paint, primer, and polishing agent, as an excellent insulator for electrical and electronics applications, and for coating seeds [24,25].

For dentistry applications, shellac is used for partial dentures and frameworks, clasps, primary crowns and bridges, full dentures, orthodontic appliances, anti-snoring devices, and various types of mouth guards and splints [26]. Shellac base plates are useful in constructing special trays and temporary denture bases [27,28]. Regardless of its color, brown shellac (the naturally occurring form), as well as the white form (bleached shellac), can be used in prosthetic dentistry as temporary baseplates for occlusal rims and trial dentures [29].

There is a distinct difference between the refining and bleaching processes of shellac. The refining process is a physical activity restricted to the purification of crude shellac from all pollutants such as sand, dirt, bark and wood particles, and insect bodies. Conversely, bleaching is a chemical process necessary to decrease the color of shellac by using a bleaching agent, such as hydrogen peroxide or sodium hypochlorite. Both processes can be achieved using the solvent extraction method, which is a gentle process that does not change the chemical structure of shellac, using different specific solvent systems and creating a product that is subsequently suitable for different dental applications [30,31,32,33,34,35]. Since the utilization of shellac has been limited because of its dark color and instability, as well as its replacement by synthetic polymers (24), it is necessary that it be bleached. Hydrogen peroxide is more environmentally friendly than sodium hypochlorite, due to the release of chlorine gas upon the oxidation reactions of the latter reagent. The changes in physicochemical properties and the whiteness of bleached shellac are dependent upon the bleaching process conditions, including pH, time, and bleaching agent volume [36,37,38,39,40,41,42].

The aims of the present work were: (a) to invent a procedure and an apparatus using microwave irradiation beams for the purification of shellac CFs into a purer and more reliable raw material that is suitable for different applications, (b) bleaching the MTS shellac floss using hydrogen peroxide, and (c) exploring suitable dental applications of the obtained bleached MTS shellac floss, based on its quality.

## 2. Materials and Methods

### 2.1. Raw Material

Commercial CFs of shellac with a light orange aspect were obtained from Jeddah market and were gently crushed to about 0.20 cm × 0.20 cm each, using a Fritsch Pulverizette, model no. 3214, Fritsch GmbH, Idar-Oberstein, Germany. This size allows the flakes to fall within the spinner cavity without dispersion in the air, and to be easily melted without combusting into carbon, leading to their conversion into floss.

### 2.2. Shellac Production Devices (Thermospinners)

Two different devices were used in the present study (see Figure 1) to carry out this task: (I) an electric thermospinner (ETS), which is a traditional candy floss machine with electric heating coils; and (II) a microwave thermospinner (MTS). Both designs (ETS and MTS) were adapted conceptually but differ in terms of mass production and heating tools, based on the centrifugation forces and thermal effect. Employing centrifugal force facilitates separating the pollutant particles from the CFs according to their size, shape, and density [10].

#### 2.2.1. Electric Thermospinner

The electric thermospinner (ETS) was the first design used to synthesize shellac floss (Figure 1A). It consists of four components: (a) the electric motor (1050 W, 220 V/50 HZ, 2300 rpm); (b) the spinner head, composed of stainless steel, containing a central cavity into which the CFs or any thermoplastic materials are inserted; (c) electric heating coils embedded within the spinner head (between its double walls); (d) a motor for rotating the spinner head up to approximately 2300 rpm, using a belt.

#### 2.2.2. Microwave Thermospinner

The microwave thermospinner (MTS) was the second design used for the synthesis of shellac floss (Figure 1B). It consists of two main units, namely, the microwave generator unit (MGU), acting as the heating tool, and the spinner head.

##### Microwave Generator Unit (MGU)

This unit consists of the following components: (a) a magnetron 2M214 39F (06B) code 2B71732E, used in LG microwave ovens (power 900 W, anode voltage 4.20 kVp, frequency 2460 MHz); (b) a high-voltage transformer (1000 E-1E, 220 V, 60 Hz); (c) a high-voltage capacitor (2100 V.AC, 1µF ± 3%, 50/60 Hz, and 10 MΩ).

The cavity of the magnetron used in this investigation is a high-powered vacuum tube, which acts as a self-exciting microwave oscillator capable of converting high-voltage electric current into microwave beams. In the magnetron, the generated crossed electron and magnetic fields are utilized to produce the high-power output essential for radar transmitters at different frequencies of 0.6–95 GHz [10,43]. Since magnetrons are usually designed to work at a fixed frequency, the magnetron used in the present invention emits a constant frequency of 2.46 GHz.

In the MTS device, the magnetron is the main and unique source for generating microwave energy for the MGU (Figure 1). The radiated microwave energy travels in the air through a stainless-steel pipe (1-inch internal diameter) and is distributed into a metal cavity by conduction. Subsequently, the spinner head temperature is tuned to approximately 75–80 °C, which is adequate to melt the shellac flakes [10].

The high-voltage transformer (HVT) is an electric power converter that maximizes the 220 V electric current into high-voltage alternating current (AC), which is then passed to the magnetron. The magnetron then converts high-voltage AC, received from the HVT, into the required 2.45 GHz [44].

The high-voltage capacitor (HVC) is connected to the magnetron and transformer via a diode to the outlet of the waveguide. It is essential to use a diode, which acts as a two-terminal electronic component that is able to conduct the current primarily in one direction (asymmetric conductance). For further illustration, the diode has low resistance in one direction, and high resistance in the other direction [10,44]. In addition, a small fan must be directed to distribute air around the transformer and the magnetron, in order to cool them. The waveguide is a short metallic device that is used to transmit the generated microwave beam from the magnetron toward the thermospinner head.

##### Thermospinning Units

Two thermospinners were used to convert the shellac flakes into shellac floss separately (Figure 1): (A) an electric thermospinner (ETS); (B) a microwave thermospinner (MTS). They differed in terms of the heating sources and the head cavity size. For the ETS, the spinning head was composed of stainless steel that contained a central cavity for inserting shellac CFs or any thermoplastic material. The MTS device consisted of the following: (a) a hemispherical spinner head composed of stainless steel (38 cm max. width × 25 cm height, and 25 cm for the upper hole) onto which the shellac flakes were placed; (b) a magnetron; (c) a motor/head-coil connector between the motor and the spinner head; (d) a high-speed motor for rotating the spinner head up to approximately 3600 rpm; (e) three coil supports for fixing the high-speed motor in a manner that absorbed the violent vibrations arising from the high speed of the axial motor.

### 2.3. Production of the Shellac Floss

Since the CFs of shellac have a non-homogeneous polymeric matrix and were contaminated with different pollutants, they were subjected to a novel refining process to obtain a more purified end product (shellac floss). Accordingly, the MTS invention in this paper aims to produce a highly homogeneous product with high-purity material, suitable for advanced nanometric applications.

The shellac floss was synthesized using two different thermospinning devices (Figure 1), namely, an electric thermospinner, which comprised an ordinary commercial candy floss machine, and a microwave thermospinner. The data collected after using both devices for shellac CFs and shellac floss were compared, to (a) confirm that neither electric coil heating nor microwave irradiation degraded the shellac quality, and (b) identify what technique could enhance the shellac quality.

The sequential steps of shellac synthesis are shown in Figure 2 and Figure 3. The CFs were inserted into the cavity of the spinner head using a directed microwave beam, radiated from a specially constructed microwave generator unit (MGU). Since shellac CFs soften at 65–70 °C and melt at 75–80 °C, it was found that when the spinner-head temperature reached the melting point of the shellac, its molten form was converted into fibers (floss) by the action of heating and the centrifugal forces that arose outside the spinner-head cavity. The floss was collected through a perforated bowl, accumulated, and then stored until use. Different physical and chemical characterizations of the shellac floss were carried out to study the shellac quality, as affected by the present invention.

### 2.4. Characterization of the Unbleached Shellac Floss

In order to confirm that the present invention enhanced the properties of the final product (ETS floss and MTS floss) compared with the shellac CFs, which have no uniform quality and contain many contaminants, the most important shellac quality parameters were characterized [3,5,19,45,46,47,48,49,50,51,52,53,54].

#### 2.4.1. Physical Properties of the Unbleached Shellac Types

Six physical properties of the three shellac types studied (CFs, ETS floss, and MTS floss) were determined, namely color index (CI), specific gravity (SG), flow of molten shellac (FSM), polymerization time (PT), moisture content (MC) and X-ray diffraction (XRD).

The CI of shellac is determined by a method described in a previous paper [38], occurring during processing or soon after. The appearance is determined by the natural coloring pigments present in lac resin. The CI was determined by comparing the color of a standard iodine solution with a clear shellac sample dissolved in ethyl alcohol (20% ethanolic solution of shellac) [35,45], using UV absorption at 430 nm (Spectrophotometer Spectronic 20, Bausch & Lomb, Rochester, NY, USA), as applied by Saengsod et al. [19].

The SG of shellac was determined by first filling a glass tube of known weight (W) with fine-ground shellac and weighing it (W_1_). Subsequently, the same glass tube was filled with the same volume of deionized water and weighed again (W_2_). The specific gravity (δ) was calculated using the following formula: δ = (W_1_ − W)/(W_2_ − W), as applied by Sharma et al. [3] and Sharma [5].

The FSM was calculated based on that described in [45] by melting a sample of shellac powder that had been passed through a 20-mesh sieve and retained on a 40-mesh screen. The test was performed in a vertical test tube for 3 min at 100 ± 1 °C in a water bath, by tilting the test tube to an angle of 15° for 12 min, to allow the sample to flow down the tube and to determine the distance in millimeters [5,35,45].

The PT of shellac can be expressed as heat-hardening per time period (150 ± 0.5 °C). The shellac needs to satisfy this test within a particular time range. The method was performed as described in [45]. The method consists of heating the shellac sample at 150 ± 1 °C and recording the time needed to attain a rubbery state, as indicated by the “spring back” of a glass rod when twisted through a full circle in the molten resin [5,8,45].

The MC of each of the three shellac types was determined according to Blanvalet et al. [55] and elsewhere [56]. Approximately 5 g of sample was spread in a thin layer within a desiccator that was freshly filled with a novel drying system containing phosphorus pentoxide crystals. The desiccator was exhausted using a vacuum pump under negative pressure of approximately 3 mmHg until a constant weight was obtained. The MC was calculated using the following equation:MC, % = [{(Wa − Wd)/Wd} × 100],
where

Wa and Wd represent the air-dried and the oven-dried weights, respectively, for a certain shellac type.

The XRD spectra of the shellac were used to investigate its crystallinity using the XRD-D2 Phaser Bruker (Bruker AXS Inc., Madison, WI, USA). The crystallinity of the shellac samples was measured by the ratio of the allocated crystalline domains of the samples to the total material, including the crystalline and amorphous parts. The generator was adjusted to 30 kV and 30 mA. Using Cu K-alpha radiation with a wavelength of 0.15418 nm, the samples were exposed for 3000 s. All the experiments were performed in the reflection mode at a scan speed of 4°/min, in steps of 0.05°. All the samples were scanned in a 2θ = 26° range, varying from 4° to 30° [19,48,51,52].

#### 2.4.2. Chemical Properties of the Unbleached Shellac Types

Five properties of the unbleached shellac forms (CFs, ETS floss, and MTS floss), namely, insolubility in hot alcohol (IHA), ash content (AC), waxiness content (WC), acid value (AV), and Fourier transform infrared spectroscopy (FTIR) were examined.

The IHA of shellac is the portion of insoluble matter that is insoluble in hot 95% ethyl alcohol, then denatured with methyl alcohol, according to Hartman [46]. Other determination methods suffer from the incomplete solubility of the shellac wax in the hot 95% alcohol, the possible loss of fine particles of insoluble matter, and the retention of some of the solvent by the filter medium. About 5 g of shellac was boiled in 100 to 120 mL of alcohol for 30 min, filtrated through a heated extraction thimble, and extracted for one hour in a specified apparatus. Finally, the thimble was oven-dried at 105 °C and weighed.

The AC of shellac was determined based on the method described in [45]. A known weight of the sample was heated on a silica or platinum crucible at a low heat, not exceeding dull redness, until all carbon was eliminated, then the crucible and its contents were finally heated in a muffle furnace at 650–750 °C for 4 h. The amount of ash was computed as a percentage, based on the parent lac [5].

The WC of shellac was determined using the method described by Stillwell [54]. Approximately 5 g of fine powdered shellac was dissolved by boiling it in 150 mL of sodium carbonate solution (2%, *w*/*w*). After cooling and suction filtration, it was washed with deionized water until colorless and, finally, washed with alcohol (70%), and subsequently oven-dried until all the solvent had evaporated. After that, the sample was extracted with carbon tetrachloride, using a Soxhlet apparatus. The solvent was distilled off and the wax was dried and weighed.

The AV of shellac was calculated using the acid–base titration method, adapted by Tang et al. [52] and applied by Saengsod et al. [19] and Farag and Leopold [57]. Approximately 0.4 g of ground shellac was dissolved in a mixture of diethyl ether and ethanol (1:1) and titrated with 0.1 M potassium hydroxide solution. Due to the dark color of the shellac solutions, instead of using a color indicator, the endpoint was determined potentiometrically.

FTIR was used to study the chemical structure of the three shellac samples (CFs and the ETS and MTS flosses) using a Bruker Tensor 37 FTIR spectrophotometer, Munich, Germany to ensure that the thermospinning did not distort the parent chemical constituents of the shellac. The samples were oven-dried at 100 °C for four hours, mixed with KBr at a ratio of 1:200 (*w*/*w*), and pressed under vacuum into pellets. The FTIR spectra of the samples were recorded in transmittance mode in the range of 500–4000 cm^−1^ [19,51].

### 2.5. Bleaching the MTS-Shellac Floss

Bleached shellac solution was prepared according to the method described by Saengsod et al. [24] with substitution of sodium hypochlorite with hydrogen peroxide. Ten grams of the MTS-shellac floss was dissolved in 100 mL of ethanol (95% *w*/*w*). Twenty milliliters of hydrogen peroxide solution (10% *w*/*v*) was mixed with the floss solution for about 2 h at ambient temperature. The precipitation process was performed by the addition of dilute sulfuric acid, followed by filtration and washing with excess water. Finally, the solid was dried at 25 °C for 2 days. The pH of the final rinse water was equal to purified water (pH 6.98).

#### 2.5.1. Preparation of the Bleached Shellac (BS) Films

A casting/solvent evaporation technique [19] was employed to prepare the bleached shellac film. About 10 g of CFs were dissolved in 100 mL of ethanol (95%, *w*/*w*) overnight and then the final concentration was adjusted to 12% *w*/*w*. Casting the shellac solution was performed on a novel non-sticky plate fabricated from poly(methyl methacrylate) (PMMA) that facilitated peeling the PS off the surface after evaporating its solvent at 50 °C for 2–3 h.

#### 2.5.2. Characterization of the BS Film

Six effective properties of the BS were investigated: namely, color change (CC), acid value (AV), insoluble solid matter (ISM), moisture content (MC), ash content (AC), and Young’s modulus.

The CC of the BS was determined by UV absorption at 430 nm. Exactly 0.1 g of shellac was dissolved in 50 mL of ethanol (95% *w*/*w*) for 3–4 h with continuous stirring and then centrifuged at 4000 rpm. Color change was calculated using the following formula:CC, % = [(C_1_ − C_2_)/C_1_ × 100], 
where C_1_ and C_2_ are the total colors of the BS before and after bleaching, respectively.

The MC of each of the BS was determined according to Blanvalet et al. [55] and in another work [56], using a method similar to that used for the MTS floss.

For the ISM, the alcoholic solution of shellac was centrifuged and then filtrated, using Whatman filter paper. The titration was performed using 0.1 N of sodium hydroxide. Due to the dark color of shellac, a pH meter instead of a color indicator was applied to detect the equivalent point. The ISM (%) of shellac on the filter paper was examined after excessive washing with ethanol (95% *w*/*w*) and was dried at 70 °C until the sample was a constant weight [19,24].

The AC of shellac was determined based on the method described in [45]. Furthermore, the AV of the BS was determined using the same acid–base titration method performed for the unbleached shellac forms (CFs, ETS floss, and MTS floss).

The E property of the shellac film produced from the MTS floss was estimated through a tensile strength test using the texture analyzer model. TA.XT Plus, Stable Micro Systems Ltd., Surrey, UK. A dumbbell-shaped film sample was prepared, of 25 mm in length and 6 mm in width, with a known thickness. An average of 5 samples was considered. The tensile strength test was applied to the shellac film strip. The texture machine was adjusted to the loading speed of 1 mm/min [19,24]. The E was calculated in MPa by the known stress (δ) and strain (ε) within the elastic limits of the stress/strain curve, according to the following equation:E = δ/ε, 
where
δ = F/A, 
where F is the force loaded to the shellac film sample and A is the sample area exposed to the referred load, and
ε = ∆L/L, 
where ∆L is the change of the sample length after loading the force and L is the original sample length.

### 2.6. Statistical Analysis

There are two experiments of complete randomized block design that were used in this investigation. The first experiment was used to detect the differences between the three shellac forms (CFs, ETS floss, and MTS floss). Furthermore, the second experiment was applied to test the differences between the bleached and unbleached MTS shellac floss. Statistical analyses of the recorded data were carried out according to the method used by Dancey and Reidy [58], using the analysis of variance (ANOVA) procedure and the least significant difference (LSD) test for the means. Significance was accepted at *p* < 0.05.

## 3. Results

Different physical and chemical properties of the unbleached shellac (CFs, ETS floss, and MTS floss) were analyzed to study the effects of the invention on shellac quality, with the results presented in Table 1 and Table 2 and in Figure 4 and Figure 5. In addition, the results of the bleached shellac film are presented in Table 3.

### 3.1. Physical Properties of the Unbleached Shellac Types

The CI values of the three shellac types were found to be 13.6%, 13.2%, and 13.4% for the CFs, ETS floss, and MTS floss, respectively (Table 1). In addition, these CI values were within the standard limits (8% to 50%) indicated by Sharma [5] for man-made shellac, machine-made shellac, dewaxed shellac, and dewaxed decolorized shellac.

All three types of shellac were found to be statistically similar in their SG. However, their SG values were calculated to be 1.183, 1.182, and 1.181, which is in agreement with the range (1.14 to 1.21) reported by both Sharma et al. [3] and in [45], as shown in Table 1.

For the FSM, it is clear from Table 1 that the MTS floss had the highest mean value (54.7 mm), followed by the ETS floss (53.3 mm). However, the FSP value of the shellac CFs was the lowest among the three shellac types (52.01 mm). It is also notable that the obtained FSM values were in the range (35 to 55 mm) referred to by Sharma [5] for his Rangeeni varieties of shellac.

The PT of shellac is an important property characterizing shellac floss. The heat-hardening of the three shellac types per time period (150 ± 0.5 °C) was determined. Table 1 reveals that the three types of shellac did not differ significantly in their PT. Their mean values were recorded as 33.81 min, 33.7 min, and 33.68 min. The resulting values are in agreement with the range (30–50 min) indicated in [45] and by Sharma [5], as can be seen in Table 1.

Table 1 reveals that the shellac CFs, ETS floss, and MTS floss contained MCs of approximately 2.08%, 1.81%, and 1.23%, respectively.

Regarding each of the three shellac X-ray diffractograms (CFs, ETS floss, and MTS floss), it can be seen that there are three essential peaks detected at the two theta angle of 11° (broad peak), 17.5° (broad peak), and 22° (relatively sharp), as shown in Figure 4.

### 3.2. Chemical Properties of the Unbleached Shellac Types

#### 3.2.1. IHA

The IHA of the three shellac types ranged from 1.1% (for the MTS floss) to 1.95% (for the CFs). However, these values were found to be within the range of 0.75–3%, as reported in [45]. The shellac samples must contain less than the specified basic limits of matter insoluble in hot alcohol, as presented in Table 1.

#### 3.2.2. AC

The AC of the shellac types (Table 1) was found to range between 0.5% and 1.0% for man-made shellac, while it should be less than 0.3% for machine-made shellac [4]. The shellac had AC values of 0.231%, 0.266%, and 0.287% for the CFs, ETS floss, and MTS floss, respectively. However, these AC mean values are accepted, according to the standard limits (< 0.3%) referred to by Sharma [5] and in [45].

#### 3.2.3. WC

The WC of shellac ranged between 2.3% and 2.71% for the three shellac types examined (Table 1). These values are comparable to those indicated by other researchers, who reported standard limits of 2.5–5.5% and 3–5% [5,45].

#### 3.2.4. AV

For the AV of shellac, it was found that all three products were statistically different, although they lay within a very narrow range (68.11–69.79 mg KOH/g for ETS floss and CFs). In addition, the MTS floss had a medium AV value. However, these values are situated within the standard limits (65–75 mg KOH/g), as referenced by Sharma et al. [3].

#### 3.2.5. FTIR

The FTIR was used to explore the chemical constituents for each of the three shellac types, namely, the CFs, ETS floss, and MTS floss. To monitor the molecular structure of the shellac after the thermospinning processes, the FTIR spectra for all three shellac types were recorded.

As shown in Table 2 and Figure 5, the FTIR spectra of all the shellac types presented vibration bands that correspond to those reported by other researchers.

The largest FTIR bands were recorded at 1252, 1713, 2856, 2933, and 2995 cm^−1^ for all shellac types, namely, CF, ETS floss, and MTS floss.

Weak signals can be observed at 883 cm^−1^ (C–H out-of-plane deformation in aldehydes) and 946 cm^−1^ (O–H out-of-plane deformation in carboxylic acids) for all three types of shellac. This can be attributed to the out-of-plane deformation of the hydroxyl groups in carboxylic acids, to the out-of-plane deformation of the C–H groups in aldehydes, or to their combination, as illustrated by Shearer [59]. Furthermore, the absorption band at 1048 cm^−1^ may be due to the stretching vibrations of the C–C bonds [60]. The absorption band at 1173 cm^−1^ may be due to the stretching vibrations of the C–O bond [60]. The O–H bending vibration was identified at 1252 cm^−1^, which is close to that indicated by Brajnicov et al. [61]. The C–O stretching vibration was found to be at 1260 cm^−1^, which is approaching those values indicated by Saengsod et al. [19] and Brajnicov et al. [61], while it is different from those found by Licchelli et al. [60].

Furthermore, the medium symmetric bending of CH_3_ arose at 1376 cm^−1^ [60]. The weak signal detected at 1414 cm^−1^ was related to the CH_2_ groups present in the ester chain. In addition, the signal at 1639 cm^−1^ corresponded to the C=C stretching vibration of vinyl [59]. The medium asymmetric bending of CH_3_ was detected at 1463 cm^−1^ [60]. The strong vibration peak at 1713 cm^−1^ can be attributed to the stretching vibration of the C=O groups found in esters, which nearly corresponds to those found by Bercea et al. [64].

The medium signal indicating the CH_2_ symmetric stretching vibration was found at wavenumbers of 2856 cm^−1^ and 2855 cm^−1^ for the present investigation and for that carried out by Ravi et al. [65].

The strong asymmetric stretching vibrations of CH_2_ can be easily observed at 2933 cm^−1^ and 2932 cm^−1^ for the present investigation and for that conducted by Ravi et al. [65]. In addition, the strong symmetric stretching vibrations of CH_2_ can also be detected at 2856 cm^−1^ and 2855 cm^−1^ for the present study and for that carried out by Ravi et al. [65], as shown in Table 2 and Figure 5.

### 3.3. Properties of the BS-Film

The important properties of the BS film, namely, CC, MC, ISM, AC, AV, and *E*, were determined and compared with their analogous unbleached samples, as presented in Table 3.

The CCs of the BS were found to be at 95.74%, which is higher than those values determined by other researchers.

Table 3 reveals that the unbleached parent MTS-SF and the bleached BS contained MCs of approximately 1.37% and 1.23%, respectively.

The ISM of the shellac was decreased from 4.58% for the unbleached MTS floss, moving down to 1.82% for the bleached sample. The ACs of the unbleached and bleached MTS-SF samples were found to range between 0.287% and 0.216%, respectively. However, these AC values are accepted, according to the standard limits (<0.3%) referred to in [5,45].

It can be seen from Table 3 that the mean values of the AV of the shellac samples were found to be 68.41% and 72.01% for unbleached and bleached MTS flosses, respectively.

The *E* value of the bleached shellac film (10.14 MPa) is statistically lower than that of the unbleached MTS shellac sample (9.88 MPa), as is clear from Table 3.

The resulting data lay in the acceptable ranges found by other researchers, using different bleached shellac types, another bleaching agent, and other methods [19,45].

## 4. Discussion

Electromagnetic waves can be emitted into the system by virtue of their crossed electric and magnetic fields (Figure 1). These fields can generate forces and moving charges in the system, and then work on them. If the electromagnetic wave frequency is equal to the natural frequencies of the system (such as microwaves at the resonant frequency of water, the stainless-steel body of the microwave thermospinner, and/or the molten polymers of the shellac molecules), the transfer of energy will occur with greater efficiency.

The stainless-steel body of the microwave thermospinner, as well as the CFs itself, are affected by microwave beams, at a frequency of 2.46 GHz and a wavelength of 12.24 cm [18,65,66,67]. It is worth mentioning that molecules of moisture, unlike the other chemical constituents of shellac, absorb energy from the microwave beams in a process called dielectric heating. Many of these molecules, especially moisture, are electric dipoles (having a positive charge at one end and a negative charge at the other). Accordingly, they are able to rotate while they try to align themselves with the alternating electric field induced by the microwave beams. This molecular alignment, as well as molecular accidents, are responsible for generating heat. Microwave heating is most efficient regarding liquid water and is much less efficient on fats and sugars (which have fewer molecular dipole moments). We surmise that the temperature generated from the stainless-steel spinner head is responsible for the shellac melting, rather than that heat expected to be generated through the highly vibrating molecules of the shellac’s chemical constituents.

One disadvantage of using microwave beams to heat the spinner head is that microwaves have hot and cold spots (Figure 1). Exploring the propagation line (the baseline) of the microwave sinusoidal curve, it can be seen that cold (damping) spots are analogous to the intersection points of both curves of the magnetic and electric waves. Fortunately, this defect was considered and eliminated in our invention since the spinner head is rotated, whereby the damp spots are continuously alternated.

For the unbleached shellac floss (Table 1), it was found that all the shellac floss properties lie within the standard limits suggested by other researchers [3,4,38]. In addition, all these data indicated that there were no changes to the floss properties examined, due to the heating tools (the electric coils and the microwave radiation beam), except for insolubility in hot alcohol, acid value, and moisture content in the case of the MTS floss. Although there were no statistical differences in the properties for almost all the properties studied, with the exception of the IHA, AV, and MC, the MTS floss exhibited superior values compared with the other types. These findings confirm that the microwave thermospinner invention did not degrade the shellac quality and, subsequently, that the suitability of the microwave thermospinner is endorsed for the purification of shellac.

### 4.1. Physical Properties of the Unbleached Shellac Types

#### 4.1.1. CI

It is clear from the CI data presented in Table 1 that both thermospun flosses had slightly lower CI values than those for the parent shellac CFs. Furthermore, the MTS floss had a higher CI value than that produced using electric thermospinning. This finding indicates that the MTS process retained the pigment content as well as the colored compounds in the synthesized shellac floss, compared with that of the ETS floss. This difference in the CI may be a result of the different origin of the parent shellac but is most likely due to thermal treatment during the refining [35] and/or flossing process.

#### 4.1.2. SG

The statistical similarity among the SG of the three types of shellac, as seen in Table 1, reveals that the choice of thermospinning tools had no influence on the SG of the shellac. Accordingly, neither the ETS nor the MTS impacted the SG of the shellac; therefore, it can be stated that improving the shellac industry by using thermospinning processes, either via ETS or MTS, is a safe and an efficient target in terms of retaining the parent SG-quality.

#### 4.1.3. FSM

When shellac is used as a thermoplastic material or is heated, its degree of fluidity is changed and it gradually loses water, with the corresponding loss of its plastic properties [35,45,68,69]. This thermoplasticity can be restored by re-heating polymerized shellac in the presence of a certain relative humidity, under certain pressure forces [69]; however, in this case, some secondary reactions certainly take place, as well as hydration, since the iodine number is increased [68].

Since the FSM is the displacement in mm achieved by moving molten shellac on a diagonal surface, tilted at an angle of 15°, the higher the FSM value, the higher the displacement achieved by the molten shellac. Based on this phenomenon, as well as the FSM data results, it was found that the MTS provided the highest FSM value for the shellac products, followed by the ETS. Accordingly, the molten MTS floss had the highest ability of thermal flow, compared with either the shellac CFs or the ETS floss. This finding can be explained by the mild heat homogeneity among the shellac matrix offered by the microwave irradiation facility.

Accordingly, the tooth varnish containing the MTS floss is expected to form a continuous film around the tooth [5,45,68] due to its high FSM and its relative slipping resistance on the tooth surfaces. Furthermore, the presence of the MTS in the varnish is believed to accelerate the equilibrium moisture in the varnish sample above that of merely the adsorbed water attracted by the free hydroxyl groups of the shellac [68].

#### 4.1.4. PT

Since the three types of shellac did not differ significantly in their HP (Table 1), it can be stated that the thermospinning process, especially the microwave type, is a suitable tool to enhance shellac purity and, subsequently, its quality.

#### 4.1.5. MC

Regarding the MC results (Table 1), the MTS floss was found to have the lowest MC value, followed by the ETS floss. However, the shellac CFs had the highest MC value. It is notable that shellac should not contain more than 2% of moisture. Accordingly, it can be a candidate for incorporation in insulating composites, as well as in some dental applications, especially tooth varnish.

#### 4.1.6. XRD

The similarity in the XRD diffractograms of the three shellac types studied, as well as the broad peaks detected, indicates that there are no distinct sharp peaks in the shellac spectrum, which confirms its amorphous state [19,52]. Considering the similarity among the three XRD diffractograms, it can be concluded that neither the thermospinning processes using electric heating coils nor the microwave irradiation beams degraded the shellac quality. Accordingly, thermospinning can be used for converting shellac flakes into flosses without changing their essential crystallographic parameters. Comparing the XRD peaks (b) and (c) for the ETS floss and the MTS floss, respectively (Figure 5), the peak found at two theta waves of 22° was slightly sharper for the MTS floss compared with that detected for the ETS floss and the shellac CFs. This latter finding points to the role of microwave irradiation beams as a heating tool in the thermospinning of shellac, in which the shellac crystallinity may be slightly enhanced.

### 4.2. Chemical Properties of the Unbleached Shellac Types

#### 4.2.1. IHA

The lowest IHA value found for the MTS floss reflects the role of the MTS process in reducing the IHA property of shellac and, subsequently, enhancing its solubility in hot alcohol, compared with those values found for the parent flakes, as well as for the ETS floss.

#### 4.2.2. AC of the Unbleached and BS-Film

There were no significant differences found among the studied ACs. Based on the AC results (Table 3) the slight increase in the AC value belonging to the MTS floss can be attributed to microwave irradiation beams providing the highest shellac purity, in which the basic weight was reduced, leading to an increase in the percentage yield of the AC. It is worth mentioning that AC is a non-desired property in the chemical industry, due to the probable interaction between its minerals and other reagents.

#### 4.2.3. WC

The WC of the MTS floss was found to be approaching that of the shellac CFs, compared with that of the ETS floss. This indicates that heating the CFs using microwave irradiation beams was very mild and offered more heating homogeneity than using electric heating coils.

#### 4.2.4. AV

There is a significant difference in AV among the three shellac products (in CFs and both the thermospun flosses). The lower AV values of the thermospun flosses (ETS and MTS) indicated that the thermospinning tools did not break the ester bonds of the shellac matrix into the free carboxyl groups [19]. As a result, the thermospun flosses did not cause the increased pH of the varnish upon contact with gum tissues.

Since shellac shows a pH-dependent solubility because of its acidic character, the dissolution properties of the investigated shellac types can be correlated with their acid values, according to the findings of Farag and Leopold [57]. In addition, they found that the aging of shellac results in a decrease in the AV and in shellac solubility.

#### 4.2.5. FTIR

FTIR was used to determine the constancy degree of the chemical composition among the three types of shellac (CF, ETS floss, and MTS floss) in order to confirm that the novel thermospinning processes, especially the MTS process, are suitable to convert CFs contaminated with different pollutants into more purified shellac floss without distorting the parent chemical quality.

It is essential to study the FTIR technique in order to strengthen the recommendation to replace the traditional machinery used for preparing shellac CFs with a more reliable, safe, and economical process, such as the MTS process based on the stability and/or enhancement of the parent chemical quality of shellac.

As can be seen in Table 2 and Figure 5, the FTIR spectra of all three shellac forms showed similar general spectral aspects. The largest FTIR bands recorded at 1252 cm^−1^, 1713 cm^−1^, 2856 cm^−1^, 2933 cm^−1^, and 2995 cm^−1^ for all the shellac types indicate a high similarity relationship among them. Accordingly, since there is no change in the molecular structure of the shellac observed using FTIR spectroscopy among the three sample types, it can be concluded that the thermospinning process, either microwaving or electro-heating, did not affect the chemical composition of the parent shellac.

### 4.3. Properties of the Bleached Shellac (BS)-Film

The brown color of the unbleached shellac forms arises from molecules containing chromophores. hydrogen peroxide, acting as an oxidizing agent, is able to whiten them by breaking the chemical bonds of chromophores, leading to the inability of the absorption of visible light [19]. This illustration is adapted to the color change percentage of the bleached shellac sample (95.74%) achieved by hydrogen peroxide. The higher amount of bleaching agent contributed to a higher efficiency in color change [29].

Based on the MC results (Table 1 and Table 3), the unbleached and bleached MTS floss had the lowest value; therefore, it is suitable for tooth varnish. Since shellac varnish films are permeable to water vapor and absorbent to moisture, the progressive adsorption of moisture with increasing humidity is in accordance with the assumption of a relatively porous structure [68]. Accordingly, it can endure deterioration in its quality or the loss of its fluidity with time under oral conditions, compared to the other forms studied of shellac. The results of Townend and Clayton [68] showed the very great effect of a small amount of adsorbed water on the fluidity, as measured by the Victor method. Attempts to bring back the fluidity of a “dead” lac—that is, one that possessed zero flow under ordinary conditions by exposing it to a high humid atmosphere—were unsuccessful.

The mean value of the ISM of the native shellac floss (4.58%) was found to be higher than that for the bleached floss (1.82%). This indicates that using hydrogen peroxide increased the solubility of some shellac components and subsequently decreased the IS for the bleached product. The lower insoluble solids were obtained when the amount of hydrogen peroxide was lower than 20 mL. A significant increase in insoluble solids was reported with higher levels of hydrogen peroxide (*p* < 0.05). The highest amount of insoluble solids was noted for 40 mL of hydrogen peroxide [29]. The higher amount of bleaching agent contributed to the higher amounts of insoluble solids [29].

The higher AV value of the bleached MTS shellac floss (72.01 mg KOH/g), compared to that for the unbleached MTS floss (68.41 mg KOH/g), can be attributed to the effect of hydrogen peroxide upon the bleaching process of shellac. The results demonstrated that the bleaching process caused a high increase in the acid value as a result of the breaking of ester bonds to free the carboxyl groups. The result was in agreement with our previous study, showing that the hydrolyzed shellac prepared by the process of alkali treatment had a high acid value compared with the native shellac [16]. It was reviewed by Farag [35] that the pharmacopeias characterize shellac only by the acid value. The European Pharmacopoeia (Ph Eur) allows a range of acid values between 65 and 80. The acid value of most dewaxed shellac types is about 70, whereas the acid value of wax-containing shellac or bleached shellac can be considerably higher, and the acid value of aged shellac may be significantly lower. The higher volumes of HOCl_2_ and NaOH caused a greater bleaching redox reaction and hydrolysis of shellac, resulting in a higher acid value. The higher the acid value, the higher the polarity contributing to the high attraction of water; hence, a high WVPC was obtained. The high level of the bleaching agent caused a higher acid value due to the high exposure to the bleaching agent, contributing to the high free carboxylic and hydroxyl groups and, hence, the high availability of polymerization [29].

Examining Table 3 revealed that the *E* of the native MTS shellac floss (10.4 MPa) is higher than that for the bleached sample (9.88 MPa). Although this difference is statistically significant, it can be considered as a minor reduction in the E, compared to that gained by other researchers using different reagent types and/or their concentrations, shellac forms, and/or procedures [19,24]. The bleaching process had a slight effect on the Young’s modulus of the shellac films, compared with the native shellac. Accordingly, hydrogen peroxide shows a slight reduction in mechanical properties, compared to the native shellac. The slight effect of the oxidizing reagent can be attributed to the best selection of the bleaching conditions concerning the concentration of the bleaching reagent and its reaction duration. The higher amount of bleaching agent contributed to the poor mechanical properties [29].

### 4.4. Suitability of the Shellac Types for Some Dental Applications

Based on the obtained results for the three unbleached shellac forms, as well as the bleached product (BS), and the effect on the well-known quality properties of dental varnish, we expect that MTS floss is suitable for incorporation into tooth varnish formulas [70,71]. This is in agreement with that indicated by Blanvalet [71], wherein well-bleached shellac and shellac wax are needed for oral compositions comprising an active component and an adhesive film-forming component for the treatment of teeth. It is worth mentioning that both unbleached and bleached shellac floss can be used in the same application but produce relatively dark-colored and white/colorless products. Of course, the latter products are preferable in the global market.

The unbleached thermospun shellac floss (ETS floss or MTS floss) demonstrated the same FTIR spectra as the native shellac. This result suggests that the molecular arrangement of the shellac was not affected by the thermospinning process, although the morphological structure was changed, to some extent, from the flake to the floss form. Accordingly, no deterioration occurred upon the conversion of CF into floss. This satisfied us that they have the same degree of safety for use in medicinal applications.

Regarding the WC result, the MTS floss exhibits more reliability for dental varnish since the WC is the second adhesive component of the varnish.

For the bleached product, adding this novel form of shellac to tooth varnish using a simple bleaching process with an eco-friendly and cheap bleaching agent (H_2_O_2_) will provide a more intensely white aspect to the treated teeth. Accordingly, the tooth varnish containing the MTS floss-based BS is expected to form a continuous film around the tooth [5,45,68], due to its high FSM of the parent MTS floss and its relative slipping resistance onto the tooth surfaces. Furthermore, the presence of the BS in the varnish is believed to accelerate the equilibrium moisture in the varnish sample above merely the adsorbed water attracted by the free hydroxyl groups of the shellac [68]. It was shown that shellac varnish films are permeable to water vapor and are also absorbent regarding moisture. These films were formed by the evaporation of a solvent and might conceivably be in a different physical condition from lac in its native state—that is, the secretion of the lac insect—or in the manufactured condition, as shellac. In this latter case, the progressive adsorption of moisture with increasing humidity is also in accordance with the assumption of a relatively porous structure. The results of Townend and Clayton [68] showed the very great effect of a small amount of adsorbed water on the fluidity, as measured by the Victor method. Attempts to bring back the fluidity of a “dead” lac—that is, one that possessed zero flow under ordinary conditions—by exposing it to a highly humid atmosphere were unsuccessful. It is well known that oral care composition in the form of tooth varnish is preferred to be substantially free of water [55,56]. Accordingly, the lower MC in shellac, as well as in its wax, results in higher stability for the formulated tooth varnish. Interestingly, shellac varnish is fairly resistant to water, although its resistance is greatly reduced over a period of time [26,70]. The results showed the very great effect of a small amount of adsorbed water on its fluidity [68]. This may improve the stability of the varnish, preventing it from wearing off from the tooth surface shortly after application.

Based on this finding, there are no alterations required for the raw material (floss) or its machinery for it to be used in known pharmaceutical applications, such as in tablet coating for time-release drugs [28,29], as well as in dentistry applications, especially in dental varnish [70,71,72,73,74]. For instance, shellac F, the experimental fluoride varnish, composed of 5% sodium fluoride (NaF), shellac, modified epoxy resin, acetone, and silica, has been largely used in the traditional procedure of tooth lac as a desensitizing agent [74].

## 5. Conclusions and Future Perspectives

Crude shellac, a thermoplastic polymer, was subjected to a novel refining process, termed microwave thermospinning, to obtain a more purified and homogeneous end product that was free of physical contaminants (shellac floss). Two different thermospinners, heated by either electric heating coils or microwave irradiation, were individually used to convert the shellac flakes into floss. The microwave-thermospun floss was bleached using hydrogen peroxide to synthesize bleached shellac films. The three unbleached shellac types (crude flakes, electro-thermospun floss, and microwave-thermospun floss) as well as the bleached films were characterized both physically and chemically.

The mean values of the properties studied for the three types of shellac were all found to be in the standard range. Although there were no statistical differences among the three shellac floss types regarding their properties, except for their insolubility in hot alcohol, acid value, and moisture content, the microwave-thermospun floss was shown to be a superior material for almost all the properties studied. Each X-ray diffractogram of the three shellac types had three essential peaks, detected at two theta waves of 11° (broad peak), 17.5° (broad peak), and 22° (relatively sharp) for flakes, and the electric- and microwave-thermospun flosses, respectively, indicating the amorphousness of the shellac.

The largest FTIR bands were recorded at 1252, 1713, 2856, 2933, and 2995 cm^−1^ for the three shellac types. Accordingly, the thermospinning process did not affect the molecular structure of the parent shellac.

The microwave thermospinner did not distort the parent shellac quality and provided a relatively higher quality shellac floss compared with that obtained using the electric thermospinner. This result allows for the construction of larger and cheaper machines that permit the mass production of shellac floss, endorsing the suitability of the invention of such applications.

Bleaching the MTS-shellac floss produced the highest CC among the other related studies, a high reduction in the ISM, due to increasing the solubility of some of the solid constituents of the shellac sample via the oxidation reactions occurring within the bleaching process itself and decreased its Young’s modulus slightly. Accordingly, it is expected that using bleached microwaved shellac floss in proper amounts can improve varnish stability at higher temperatures. In addition, the formulated tooth varnish is expected to possess better adhesive ability, be more easily applied to teeth, result in less coloring of the teeth surfaces, and retain the activity of the active component (fluoride ions) compared with those tooth varnishes that contain ordinary shellac flakes.

In the near future, floss may become the most commonly handled type of shellac, instead of flakes or the other types of shellac in the market, due to its high purity and ease of use, and its low cost of manufacturing and transportation. Importantly, morphological, mechanical, and cytocompatibility characterization techniques must be studied and employed to show the applicability of these materials. The encapsulation of the nanoforms of drugs in shellac fibers using improvised electrospinning techniques must be extended to microwave-thermospun floss. Determining the best formulations of shellac composites and their mechanical properties is essential in the dental and pharmaceutical fields. In addition, there is a need to develop dental varnish compositions that do not adversely affect the color of teeth and that increase tooth stability.

## 6. Patent

System and method for manufacturing shellac floss (US Patent no. 11060208 B1).

## Figures and Tables

**Figure 1 polymers-15-00142-f001:**
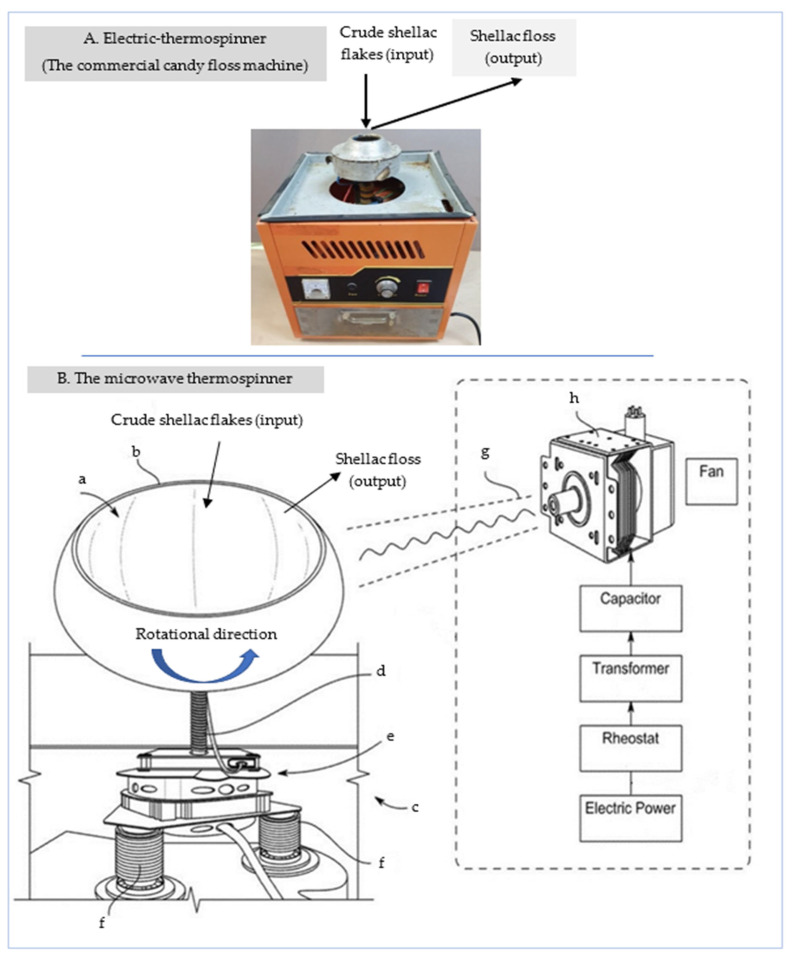
The thermospinners used to convert shellac flakes into shellac floss: (**A**) electric thermospinner (ETS); (**B**) microwave thermospinner (MTS). Components: (a) head cavity to accommodate the crude shellac flakes; (b) rotatable head; (c) kinetic transfer unit; (d) coupling shaft between the motor and the spinner head; (d) motor; (f) coil support; (g) waveguide used to transfer the generated microwave beam; (h) magnetron.

**Figure 2 polymers-15-00142-f002:**
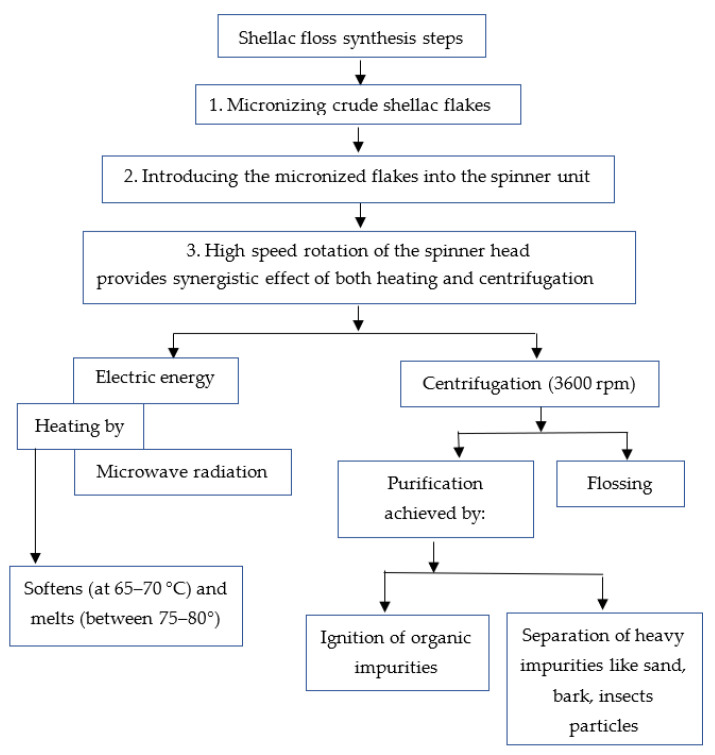
Schematic flow diagram of the manufacturing processes for the purified shellac floss.

**Figure 3 polymers-15-00142-f003:**
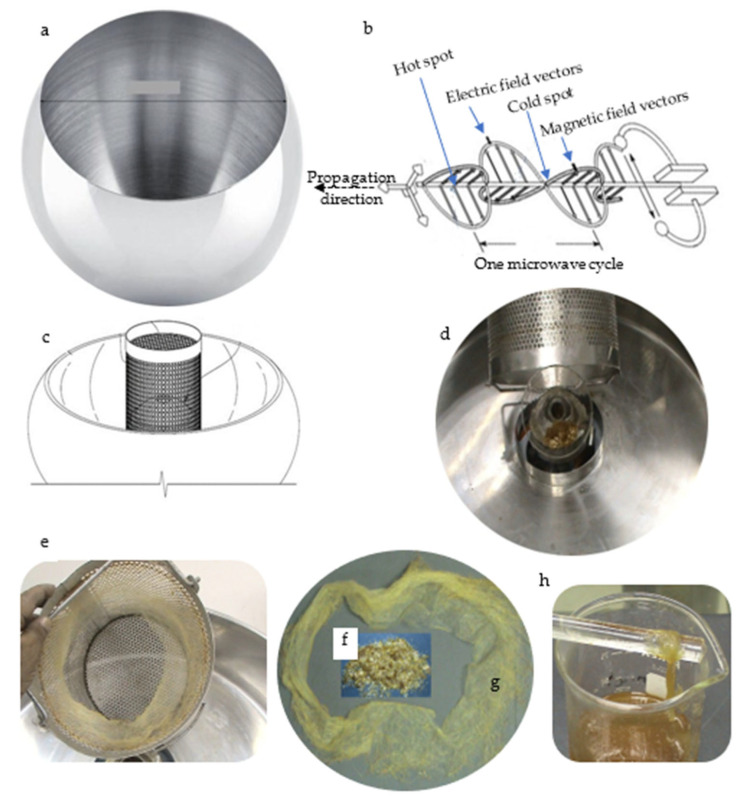
The spinner head of the microwave thermospinner (MTS) used for the production of the shellac floss: (**a**) upper cavity hole; (**b**) the electromagnetic nature of the microwave radiation; (**c**) cavity of the centrifuge spinner head to be filled with shellac flakes; (**d**) perforated bowl facing the spinner head for collecting shellac floss; (**e**) bowl for containing the collected floss; (**f**) crude flakes; (**g**) shellac floss; (**h**) shellac paste that can be used for either coating purposes or casting membranes.

**Figure 4 polymers-15-00142-f004:**
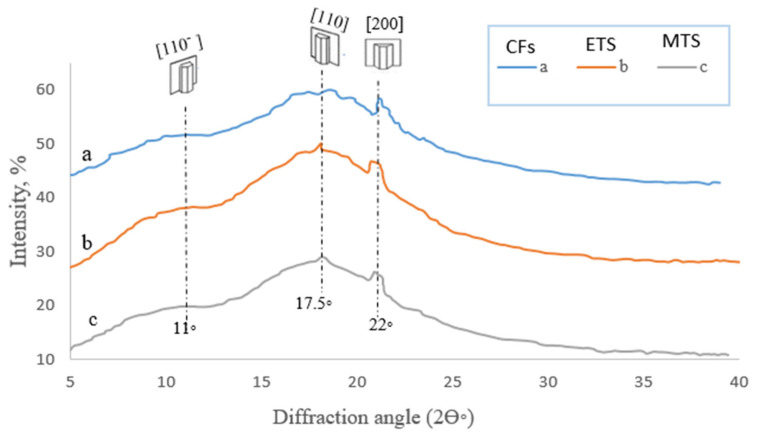
X-ray diffractograms of shellac: (a) crude flakes (CFs); (b) electro-thermospun (ETS) floss; (c) mi-crowave-thermospun (MTS) floss, according to Hindi et al. (2021) [10].

**Figure 5 polymers-15-00142-f005:**
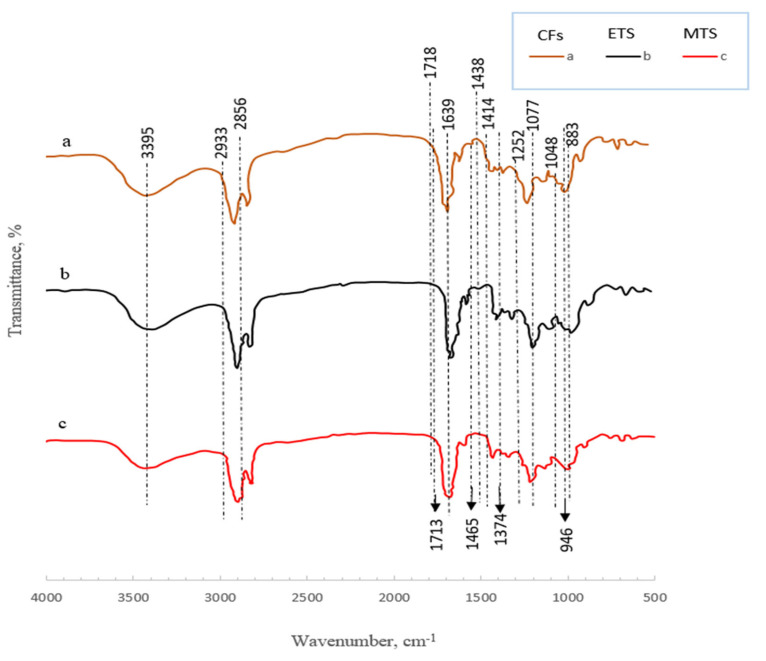
Spectrograms of FTIR shellac: (a) crude flakes; (b) electric-thermospun (ETS) floss; (c) the invented microwave-thermospun (MTS) floss, according to Hindi et al. (2021) [10].

**Table 1 polymers-15-00142-t001:** Mean values ^1,2^ of the physical and chemical properties of the three unbleached shellac types, namely, the crude flakes (CFs), electro-thermospun (ETS) floss, and microwave-thermospun (MTS) floss, compared with the standard limits.

Property		Property Value of Shellac			Standard Limits		
		CF	ETS Floss	MTS Floss	[3] ^3^	[5] ^3^	[45] ^3^
Physical	Color index (CI)	13.6 [0.41]	13.2 [0.31]	13.4 [0.29]	n.d. ^4^	8 to 50	n.d. ^4^
	Specific gravity (SG)	1.183 [0.098]	1.182 [0.044]	1.179 [0.034]	1.14 to 1.21	n.d. ^4^	n.d. ^4^
	Flow of molten shellac (FSM), mm	52.01 [1.61]	53.3 [1.93]	54.7 [1.77]	n.d. ^4^	35 to 55	n.d. ^4^
	Polymerization time (PT), min	33.81 [1.64]	33.70 [1.36]	33.68 [1.17]	n.d. ^4^	30 to 70	n.d. ^4^
	Moisture content (MC), %	2.08 [0.05]	1.81 [0.04]	1.23 [0.04]	n.d. ^4^	n.d. ^4^	≥2
Chemical	Insolubility in hot alcohol (IHA), %	1.95 [0.113]	1.8 [0.104]	1.1 [0.124]	n.d. ^4^	n.d. ^4^	0.75–3
	Ash content (AC), %	0.231 [0.092]	0.266 [0.064]	0.287 [0.061]	n.d. ^4^	n.d. ^4^	<0.3
	Waxiness content (WC), %	2.71 [0.033]	2.3 [0.042]	2.6 [0.28]	n.d. ^4^	3–5	2.5–5.5%
	Acid value (AV), mg KOH/g	69.79 [1.67]	68.11 [1.338]	68.41 [1.52]	n.d. ^4^	n.d. ^4^	65–75

^1^ Each value is an average of 5 samples. ^2^ Values within parentheses are standard deviations. ^3^ Reference. ^4^ Not defined.

**Table 2 polymers-15-00142-t002:** The FTIR bands detected for the three shellac types studied.

Related Cause	Wavelength Value, cm^−1^		Reference
	Present Study	Literature Values	
C–H out-of-plane deformation in aldehydes	883	882	[59]
O–H out-of-plane deformation in carboxylic acids	946	944	[59]
C–C stretching vibration	1048	1050	[60]
Stretching vibrations of C–O bond	1177	1175	[60]
O–H bending vibration	1252	1250	[61]
C–O stretching vibration	1260	1255	[19]
		1255	[61]
CH_3_ symmetric bending vibration	1374	1376	[60]
CH_2_ group attached to the ester chain	1414	1412	[62]
Asymmetric bending of CH_3_	1465	1463	[62]
CH_3_ asymmetric bending vibration	1438	1436	[60]
C=C stretching vibration of vinyl	1639	1636	[63]
C=O stretching vibration of esters	1713	1712	[64]
C=O stretching vibration	1718	1716	[19]
CH_2_ symmetric stretching vibration	2856	2855	[65]
CH_2_ asymmetric stretching vibration	2933	2932	[65]
O–H stretching vibration band	3395	3400 3100–3700, with the maximum at 3420	[19]
			[61]

**Table 3 polymers-15-00142-t003:** Mean values ^1,2^ of the important properties of the unbleached and bleached shellac (BS) film produced from microwave-thermospun (MTS) shellac floss (SF), namely, color change (CC), acid value (AV), insoluble solid matter (ISM), moisture content (MC), ash content (AC), and Young’s modulus (*E*).

Property	Unbleached Parent MTS-SF	BS-Film
CC, % MC, %	0 1.37 [0.04]	95.74 [1.27] 1.23 [0.08]
ISM, % AC, % AV, mg KOH/g *E*, MPa	4.58 [0.46] 0.287 [0.061] 68.41 [1.52] 10.14 [0.61]	1.82 [0.22] 0.216 [0.083] 72.01 [2.47] 9.88 [0.61]

^1^ Each value is an average of 5 samples. ^2^ Values within parentheses are standard deviations.

## Data Availability

The supporting data for the reported results, including a link to the publicly archived datasets analyzed or generated during the study, can be found under the following patent: US Patent for System and Method for Manufacturing Shellac Floss Patent (Patent # 11,060,208 issued 13 July 2021)—Justia Patents Search https://patents.justia.com/patent/11060208 (accessed on 17 October 2021).

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
