# Peer review of "Bioplastic Floss of a Novel Microwave-Thermospun Shellac: Synthesis and Bleaching for Some Dental Applications"

_polymers, 2022, doi:10.3390/polym15010142_

Round 1

Reviewer 1 Report (Previous Reviewer 2)

The authors have diligently worked to further improve their manuscript.  It forms a useful description of the thermospinning approaches to shellac flake modification which they have investigated.  I have no additional feedback suggestions.

Reviewer 2 Report (Previous Reviewer 1)

After reading the revised version of the manuscript, my opinion is the same as previously. I consider it does not fulfill the requirements to be published in a high-impact factor journal like Polymers. From my point of view, the figures and the text do not provide innovative and attractive information for the readers.

This manuscript is a resubmission of an earlier submission. The following is a list of the peer review reports and author responses from that submission.

Round 1

Reviewer 1 Report

The manuscript authored by Sherif S. Hindi et al. reports the fabrication of shellac flosses employing two different techniques, electric thermospinner (ETS) and microwaved thermospinner (MTS) for potential application in industry. After a careful revision of the manuscript, I consider the topic of the work does not meet the quality and novelty required by Polymers and it is not accurate for publication in this Journal.

The structure of the manuscript is not attractive for the readers and the English should be strongly revised. Besides, authors have not paid attention to the format and editing steps, for example, the instructions for authors included in the original template have not been removed, as can be checked in the page 4 section 2.2.2, the references section does not follow the style of Polymers. Apart from that, the multiple subsections in Results and Discussion sections do not allow to stablish a connection between techniques and make the manuscript difficult to read. Figures 1 and 7 do not provide any novel information, they are redundant, and Figures 2 and 3 are not visual and they are meaningless to highlight any novelty of the work. Importantly, morphological, mechanical and cytocompatibility characterization techniques have not been employed to show the applicability of those materials.

Author Response

The authors of this manuscript appreciate the reviewer for the constructive and positive comments. The detailed and categorically reply to each comment is as follows:

Point 1: I consider the topic of the work does not meet the quality and novelty required by Polymers and it is not accurate for publication in this Journal.

Response 1: The raised aspect and feedback is of fair concern. Perhaps, this was due to inappropriate title caption. Therefore, the authors propose change in title to reflect the true essence of the novelty and to meet with journal aims and scope. Hence, the old title “System and method for manufacturing purified shellac floss for advanced applications” is changed to “A novel microwaved-thermospinng technique to produce purified shellac floss”.

Point 2: The structure of the manuscript is not attractive for the readers.

Response 2: The authors took this feedback on serious note. To address this, the structure of the manuscript is improved by re-dividing and/or grouping each section such as Materials and Methods, Results and Discussion in a more appropriate and rational way. Few of such aspects are enlisted below:

  • For the Materials and Methods section, the heading “Characterization of the Shellac Floss” (numbered as 2.4.) is divided into two, namely physical properties of the shellac types (numbered as 2.4.1.) and chemical properties of the shellac types (numbered as 2.4.2.). Furthermore, each of the above-mentioned sub-headings was divided into sub-sub-titles as follow:
    • Physical properties of the shellac types included five properties, namely the color index, CI (2.4.1.1.), the specific gravity, SG (2.4.1.2.), the flow of shellac molten, FSM (2.4.1.3.), the heat of polymerization, HP (2.4.1.4.), and the X-ray diffraction, XRD (2.4.1.5.).
    • Chemical properties of the shellac types included seven properties, namely the insolubility in hot alcohol, IHA (2.4.2.1.), the ash content, AC (2.4.2.2.), the waxiness content, WC (2.4.2.3.), the acid value, AV (2.4.2.4.), , and Fourier transform infrared spectroscopy, FTIR (2.4.2.5.).

In addition, same modified structure of the manuscript was repeated in the Results and Discussion sections as follow:

  • For the Result section, the heading (numbered as 3) is divided into two, namely physical properties of the shellac types (numbered as 3.1.) and chemical properties of the shellac types (numbered as 3.2.). Furthermore, each of the above-mentioned sub-headings was divided into sub-sub-titles as follow:
    • Physical properties of the shellac types included five properties, namely the color index, CI (3.1.1.), the specific gravity, SG (3.1.2.), the flow of shellac molten, FSM (3.1.3.), the heat of polymerization, HP (3.1.4.), and the X-ray diffraction, XRD (3.1.5.).
    • Chemical properties of the shellac types included five properties, namely the insolubility in hot alcohol, IHA (3.2.1.), the ash content, AC (3.2.2.), the waxiness content, WC (3.2.3.), the acid value, AV (3.2.4.), , and Fourier transform infrared spectroscopy, FTIR (3.2.5.).
  • For the Discussion section (numbered as 4) is divided into two, namely physical properties of the shellac types (numbered as 4.1.) and chemical properties of the shellac types (numbered as 4.2.). Furthermore, each of the above-mentioned sub-headings was divided into sub-sub-titles as follow:
    • Physical properties of the shellac types included five properties, namely the color index, CI (4.1.1.), the specific gravity, SG (4.1.2.), the flow of shellac molten, FSM (4.1.3.), the heat of polymerization, HP (4.1.4.), and the X-ray diffraction, XRD (4.1.5.).
    • Chemical properties of the shellac types included seven properties, namely the insolubility in hot alcohol, IHA (4.2.1.), the ash content, AC (4.2.2.), the waxiness content, WC (4.2.3.), the acid value, AV (4.2.4.), , and Fourier transform infrared spectroscopy, FTIR (4.2.5.).

Point 3: The English should be strongly revised.

Response 3: The authors has the revised manuscript by a native ENGLISH. The authors believe that the revised manuscript is in good improved shape now and is up to the journal publication standards.

Point 4: Authors have not paid attention to the format and editing steps, for example, the instructions for authors included in the original template have not been removed, as can be checked in the page 4 section 2.2.2.

Response 4: The instructions for authors included in the original template [from line 158-172] is now removed. In addition, the revised manuscript is aligned to the journal policy and guideline.

Point 5: The references section does not follow the style of Polymers.

Response 5: All the references are corrected to follow the reference style of Polymers such as:

  • Publishing years is in bold format.
  • Author’s names in each citation is separated by semicolons.
  • Names of the journals were corrected to fulfill the abbreviation style.

Point 6: The multiple subsections in Results and Discussion sections do not allow to stablish a connection between techniques and make the manuscript difficult to read.

Response 6. The authors have treated the problem of the multiple subsections in Results and Discussion sections as illustrated in Point 2 as follow:

  • For the Result section, the heading (numbered as 3) is divided into two, namely physical properties of the shellac types (numbered as 3.1.) and chemical properties of the shellac types (numbered as 3.2.). Furthermore, each of the above-mentioned sub-headings was divided into sub-sub-titles as follow:
    • Physical properties of the shellac types included five properties, namely the color index, CI (3.1.1.), the specific gravity, SG (3.1.2.), the flow of shellac molten, FSM (3.1.3.), the heat of polymerization, HP (3.1.4.), and the X-ray diffraction, XRD (3.1.5.).
    • Chemical properties of the shellac types included five properties, namely the insolubility in hot alcohol, IHA (3.2.1.), the ash content, AC (3.2.2.), the waxiness content, WC (3.2.3.), the acid value, AV (3.2.4.), , and Fourier transform infrared spectroscopy, FTIR (3.2.5.).
  • For the Discussion section (numbered as 4) is divided into two, namely physical properties of the shellac types (numbered as 4.1.) and chemical properties of the shellac types (numbered as 4.2.). Furthermore, each of the above-mentioned sub-headings was divided into sub-sub-titles as follow:
    • Physical properties of the shellac types included five properties, namely the color index, CI (4.1.1.), the specific gravity, SG (4.1.2.), the flow of shellac molten, FSM (4.1.3.), the heat of polymerization, HP (4.1.4.), and the X-ray diffraction, XRD (4.1.5.).
    • Chemical properties of the shellac types included seven properties, namely the insolubility in hot alcohol, IHA (4.2.1.), the ash content, AC (4.2.2.), the waxiness content, WC (4.2.3.), the acid value, AV (4.2.4.), , and Fourier transform infrared spectroscopy, FTIR (4.2.5.).

Point 7. Figures 1 and 7 do not provide any novel information, they are redundant

Response 7. We assure that Figure 1 is a core stone of the present investigation since it presents the principle idea of the manufacturing process of the shellac floss as well as the components of the thermospinners (electric and microwaved) used for converting the shellac flakes into the floss. Accordingly, this Figure was kept.

We confound the parent three. The authors strongly believe that Figure 1 contains vital and core information, since it presents the principle idea of the manufacturing process of the shellac floss as well as the components of the thermospinners (electric and microwaved) used for converting the shellac flakes into the floss. 

For Figure 7, the authors took this feedback seriously by confounding the three parent sub-figures (7a, 7b, and 7c) to prevent any duplication of information. In addition, the authors believe that this figure truly reflects the illustration of the microwave behavior and its effect on the shellac softening, melting and flossing.

Accordingly, the authors didn’t discard these figures (1 and 7).

Point 8. Figures 2 and 3 are not visual and they are meaningless to highlight any novelty of the work.

Response 8. Figures 2 and 3 were confound together into a new Figure “Figure 2” to prevent any duplication of information. By making this improvement, the newly added Figure 2 provides more highlighting the novelty of the work.

Point 9. Importantly, morphological, mechanical and cytocompatibility characterization techniques have not been employed to show the applicability of those materials.

Response 9. Although the authors appreciate this notion suggested by the respected reviewer, they still believe that the performed properties of the three types of shellac are adequate to protrude the effectiveness of the thermospinners on the produced high quality-shellac floss and for proving that some essential parent properties didn’t distorted, while the others were enhanced. On the other hand, respecting to the reviewer’s notion, the authors modified the conclusions title from “Conclusions” to “Conclusions and Future Perspectives” which contains an added paragraph concerning to the future perspectives of the invented shellac floss.

Point 10. The introduction must be improved to provide sufficient background and include all relevant references.

Response 10: The introduction section is improved to provide sufficient background and include all relevant references. This improvement was done as shown in the attached corrected manuscript.

Point 11: The cited references relevant to the research must be improved.

Response 11: The cited references relevant to the research is improved by adding additional references to offer additional information related to methods (Materials and Methods section) as well as for showing accuracy of the obtained data (in Results section), confirming our findings (in Discussion section).

In addition, a new reference belonging to statistics was added as no. 50 in the reference section “[50] Dancey, C. P., and Reidy, J. Statistics without Math for Psychology (4th ed.), Pearson Education Limited, Essex, Pearson Education Limited, England, 2007” for referring to the statistical design “title no. 2.5”.

Point 12: The research design must be improved.

Response 12. For the statistical design, an individual paragraph concerning it to was added at the end of material and methods bearing no. “2.5”. The factors, and their levels used to achieve this study were shown clearly.

This added paragraph is “A complete randomized block design with three replications was used in this investigation. In addition, statistical analyses of the recorded data were done according to Dancey and Reidy [50] using the analysis of variance (ANOVA) procedure and least significant difference test (LSD) of the means. Significance was accepted at P < 0.05.

Point 13. The methods description must be improved.

Response 13. The methods description was improved by adding some additional details as shown in the attached corrected manuscript.

Point 14. The results presentation must be improved.

Response 14. The results presentation was improved as follow:

For Table 1 several modifications were done to enhancement results presentation:

  1. Its title was changed from “Mean values of the three shellac types properties produced by the electro- and micro-waved-thermospinners as compared with standard limits.” to “Mean values of physical and chemical properties of the three shellac types properties produced by the electro- and microwaved-thermospinners as compared with standard limits”. This change was performed according to the enhancement of the research design.
  2. A sub-column was added representing physical and chemical properties within the property column.
  3. Two consequent properties, namely solubility in cold water (SCW), and volatile matter content (VMC) were deleted due to they are less importance than the other properties studied. Another reason for this deletion is that their text was removed previously.
  4. The properties were rearranged in an identified constant sequence. Their sequence were changed from “Color index (CI), insolubility in hot alcohol (IHA), ash content (AC), waxiness content (WC), acid value (AV), flow of shellac molten (FSM), heat of polymerization (HP)” to “Color index (CI), specific gravity (SG), flow of shellac molten (FSM), heat of polymerization (HP), insolubility in hot alcohol (IHA), ash content (AC), waxiness content (WC), and acid value (AV).

Point 15. The conclusions must be improved to be highly-supported by the results.

Response 15. The conclusions were improved to be highly-supported by the results by adding a paragraph indicating the main clear results as shown in the attached corrected and revised manuscript.

Reviewer 2 Report

Authors have described the creation of thermospun fibrous polymer material, potentially suitable for pharmaceutical and medical applications using shellac, including transforming crude shellac into floss comprised of purified material.  The work discloses a proprietary polymer fibre thermospinning equipment developed by the authors.

The description of the spinning equipment and the characterization of the nature of the thermospun shellac is of interest, but the manuscript requires some attention by the authors before it can be considered further for publication.

The following points should be considered. 

Title and keywords:  the authors talk of use of the material in "advanced applications" and in "dental applications". These are not described in any detail in the text of the paper, or even indicated as to what the applications might be. This should aspect of the manuscript must be developed further to keep current title and keywords.

Page 1, line 29 - page 3 line 110: the authors go into great detail about the sources of shellac polymer and how if is harvested purified and provided as an article of commerce. This currently extensive section should be significantly condensed, to refer to references for the details the origins and nature of shellac.   Page 1 lines 30-35 can be retained, then Page 2 lines 85-96 (first sentence  of lines 95-96 is useful to prospective readers of the article. From page 2 line 96 (from "Furthermore, applications of shellac...") to page 3 line 105 are probably not essential.   Keep text from page 3 line 106, starting "For dentistry..."  to page 3 line 129.

Page 3 line 141: Should the title be "The shellac floss production devices"?

Page 4 lines 160- 174: should be deleted. They are instruction from the manuscript template.

Page 7 line 239:  is it heating and centrifugal force , not heating or centrifugal force? How does heating alone created fibres?

Page 11 lines 376 - 379:  you might comment that the XRD  patterns indicate all of the shellac materials studied are essentially  non-crystalline.

Page 19: authors have not included any discussion of the advanced applications or the dental applications of the  thermospun shellac fibres. This is promised in the title and the keywords. It must be added  into the discussion, otherwise  prospective readers are being misled as to the manuscript content.

Author Response

Point 1: Moderate English changes is required.

Response 1: Linguistic revision and improvement were done along with all the article divisions.

Point 2: The research design can be improved as shown in the corrected manuscript.

Response 2: An individual paragraph concerning to statistical design was added at the end of material and methods the factors and their levels used to achieve this study.

Point 3: The methods description can be improved as shown in the corrected manuscript.

Response 3: The methods description was improved by adding some additional details.

Point 4: the results presentation can be improved.

Response 4: We assure that Figure 1 is a core stone of the present investigation since it presents the principle idea of the manufacturing process of the shellac floss as well as the components of the thermospinners (electric and microwaved) used for converting the shellac flakes into the floss. Accordingly, this Figure was kept.

For the Figure 7, we confound its parent three sections into integrated one with maintaining the meant information to prevent any repetition. We can’t discard this Figure due to its importance in illustration of the microwave behavior and its effect on the shellac softening, melting and flossing synergizing the centrifugal effect.

Point 5: Conclusions supported by the results must be improved.

Response 5: Conclusions supported by the results was improved.

Point 6: The manuscript requires some attention by the authors before it can be considered further for publication.

Response 6.

Point 7. The authors talk of use of the material in "advanced applications" and in "dental applications". These are not described in any detail in the text of the paper, or even indicated as to what the applications might be. This should aspect of the manuscript must be developed further to keep current title and keywords.

Response 7.

Different advanced applications, especially in dental sector were clearly overviewed in the introduction and discussion sectors.

Point 8.

Page 1, line 29 - page 3 line 110: the authors go into great detail about the sources of shellac polymer and how it is harvested purified and provided as an article of commerce. This currently extensive section should be significantly condensed, to refer to references for the details the origins and nature of shellac.

Response 8.

This section was significantly condensed as shown in the corrected version of the manuscript.

Point 9.

From page 2 line 96 (from "Furthermore, applications of shellac...") to page 3 line 105 are probably not essential.   Keep text from page 3 line 106, starting "For dentistry...” to page 3 line 129.

Response 9. At the beginning of line 96 (page 2) the paragraph started by” Furthermore, for wood treatment,..”, it was severely condensed as  follow:

“Furthermore, shellac has been using in wood industry as a primer and polishing agent. For the electrical and electronics applications, shellac is the best natural insulator for wires and electronic units In addition, it is important to enhance printing inks properties. Furthermore, shellac is essential component in cosmetics such as a binder for mascara, shampoo, film former for hairspray, and micro-encapsulation of fragrances. In agricultural uses, it is suitable for seeds coating and leather finishes. In addition, chemical dye-stuffs industry utilizes it for micro-encapsulation technique”.

The text starting from "For dentistry...” to page 3 line 129 was kept as it is as the reviewer recommended.

Point 10.

Page 3 line 141: Should the title be "The shellac floss production devices"?

Response 10.

The title was changed to be “Microwaved-thermospinng device for manufacturing purified shellac floss”.

Point 11.

Page 4 lines 160- 174: should be deleted. They are instruction from the manuscript template.

Response 11. Lines 160- 174 (page 4) which are instructions from the manuscript template were deleted.

Point 12.

Page 7 line 239:  is it heating and centrifugal force, not heating or centrifugal force? How does heating alone created fibres?

Response 12. Investigating lines 236-238, “its molten was converted into fibers (floss) by the action of either heating or centrifugal forces arisen towards outside the spinner head cavity”, the statement was changed from “either heating or centrifugal forces” to “heating and centrifugal forces like as suggested by the respected reviewer.

Point 13.

Page 11 lines 376 - 379:  you might comment that the XRD patterns indicate all of the shellac materials studied are essentially non-crystalline.

Response 13. Regarding to page 17, lines 481-472 “as well as their broaden peaks detected, it can be indicated that there are no distinct sharp peaks in the spectrum of shellac which confirms its amorphous state”, we added that “This means that the XRD patterns indicate all of the shellac materials studied are essentially non-crystalline”.

Point 14.

Page 19: authors have not included any discussion of the advanced applications or the dental applications of the thermospun shellac fibers. This is promised in the title and the keywords. It must be added into the discussion, otherwise prospective readers are being misled as to the manuscript content.

Response 14.

The advanced applications and the dental applications of the thermospun shellac fibers were added into the discussion as shown in the corrected version of the manuscript.

Round 2

Reviewer 1 Report

I appreciate the effort made by the authors in this revised version, but unfortunately I consider this work does not meet the quality required to be published in Polymers (Q1). The topic, the methods and characterization techniques, as well as the presentation and discussion of the results do not provide significant content for the readers of this journal. I would recommend the submission to the journal Polysaccharides.

Author Response

Response to Reviewer 1 Comments (round 2)

The authors of this manuscript appreciate the respected reviewer 1 for his kind appreciation our effort in this revised version (round 1) as well as the constructive and positive comments.

The detailed and categorically reply to each comment is as follows:

The topic, the methods and characterization techniques, as well as the presentation and discussion of the results do not provide significant content for the readers of this journal.

Point 1: He consider the topic of the work does not provide significant content for the readers of this journal.

Response 1: The manuscript’s title was changed to be as follow: ‘A novel microwave thermospun shellac floss for dental varnish”.

Point 2: He consider the methods of the work does not provide significant content for the readers of this journal.

Response 2:

Point 3: He consider the presentation of the work does not provide significant content for the readers of this journal.

Response 3:

Point 4: He consider the results of the work does not provide significant content for the readers of this journal.

Response 4:

For Figure 1:

The referring box was edited to be horizontally instead of vertically.

Part of Figure 1

For Figure 3;

corrected from “f” to be “g’

Part of Figure 3

For Figure 4, The mistakes in the above xrd-difractogram were corrected in the figure below.

Figure 4. X-ray diffractograms of shellac: a) crude flakes, b) electro-thermospun floss and c) microwaved-thermospun floss.

50

100

200

150

400

350

300

250

2 Ө0

a

b

c

Intensity (counts)

22⁰

11⁰

17.5⁰

Figure 4. X-ray diffractograms of shellac: a) crude flakes, b) electro-thermospun floss and c) microwaved-thermospun floss.

Part of figure 5

For the Figure 5, the c denoting to the crude shellac flakes were added to the figure.

Point 5: He consider the discussion of the work does not provide significant content for the readers of this journal.

Response 5: The advanced applications and the dental applications of the thermospun shellac fibers were added into the discussion as shown in the corrected version of the manuscript. Furthermore, the shellac properties were correlated to the different dentistry applications, especially dental varnish.

  1. The improved discussion section

……………………………………………………………………………………………………………………….

Based on our obtained results for shellac and on the well-known quality properties of dental varnish, we expect that MTS floss is expected to be suitable for incorporation into tooth varnish formulas.

4.1. Physical Properties of the Shellac Types

4.1.1. CI

…………… Accordingly, adding such shellac to tooth varnish requires a simple bleaching process using a suitable cheap beaching agent in order to provide a white aspect to the treated teeth.

4.1.2. SG

…………………………

4.1.3. FSM

……………………….

4.1.4. HP

……………………….

4.1.5. MC

………………….. Accordingly, the lower MC in shellac, as well as in its wax, results in a higher stability for formulated tooth varnish. Interestingly, shellac varnish is fairly resistant to water although its resistance is greatly reduced over a period of time [31, 60].

4.1.6. XRD

………………….. This later finding points to the role of microwave irradiation beams as a heating tool in the thermospinning of shellac in which the shellac crystallinity may be slightly enhanced. This may improve the stability of the varnish, preventing it from wearing off from the tooth surface shortly after application.

4.2. Chemical Properties of the Shellac Types

4.2.1. IHA

The lowest IHA value found for the MTS floss reflects the role of the MTS process in reducing the IHA property of shellac and, subsequently, enhancing its solubility in hot alcohol compared with those values found for the parent flakes, as well as for the ETS floss. Accordingly, more homogeneous liquor can be obtained from the floss, which is essential for dental varnish.

4.2.2. AC

There were no significant differences found among the studied ACs. Based on the AC results, the highest AC value belonging to the MTS floss can be attributed to microwave irradiation beams providing the highest shellac purity, in which the basic weight was reduced, leading to an increase in the percentage yield of the AC. It is worth for mentioning that the AC is a non-desired property in the chemical industry, due to the probable interaction between its minerals and other reagents. Accordingly, the lower AC values of the three shellac types permit it to be utilized safely in dentistry constructions.

4.2.3. WC

The WC of the MTS floss was found to be approaching that of the shellac CFs, compared with that of the ETS floss. This indicates that heating the CFs using microwave irradiation beams was very mild and offered more heating homogeneity than using electric heating coils. Based on this finding, the MTS floss exhibits more reliability for dental varnish since the WC is the second adhesive component of the varnish.

4.2.4. AV

…………………………..

Since shellac shows a pH-dependent solubility because of its acidic character, the dissolution properties of the investigated shellac types can be correlated with their acid values according to the findings of Farag and Leopold [51]. In addition, they found that the aging of shellac results in a decrease in the AV and in shellac solubility. However, their indication that the extent of this change in physicochemical properties depends on the type of shellac, its origin and the type of refining process, as well as flossing preparation, can be taken in consideration for keeping such a varnish treatment on the teeth for as long as possible. In conclusion, it can be recommended for patients who are struggling with sensitivity and/or tooth decay.

4.2.5. FTIR

FTIR was used to determine the constancy degree of the chemical composition among the three types …………………………….Based on this finding, there are no alterations required for the raw material (floss) or its machinery to be used in known pharmaceutical applications, such as in tablet coating for time-release drugs [28, 29], as well as in dentistry applications, especially dental varnish [60,62,63].

Point 6: He consider that the introduction must be improved in order to provide sufficient background including all relevant references.

Response 6: The introduction section was condensed and english-edited in the MDPI-editing office (the editing certificate is presented at the beginning of this response report).

The old introduction

Shellac is known as a polymeric resin secreted by the female of shellac bug (Lac-cifer lacca). It is hosted by certain trees, especially Schleichera oleosa, Schleichera oleos, Ziziphus mauritania and Butea monosperma. The immature nymphs exude a resin/wax mixture through their bodies after they suck the tree’s sap, forming a defensive cell of a cocoon [1]. The shellac has been processing and is sold as dry flakes and still has been occupying its important position among the natural resins [2-4].

Shellac manufacturing involves the following processes: a) tree pruning, b) inoculation to introduce lac insects onto their suitable host plants, c) removal of phunki lac, d) harvesting which is the collecting of mature bearing from the tree, and e) shellac scraping which is separating lac crusts from lac sticks [4].  Shellac is scraped off its-bearing branches (raw shellac) using a knife to collect impure crude shellac termed as scraped shellac or sticklac with different impurities like sand, dirt, and stick pieces [5].

The purification process of seedlac is done by washing crude shellac or sticklac to remove different impurities, followed by drying process. This semi-refined type is known to be seedlac can be obtained after crushing sticklac, washing, filtration, and drying. After that, the pure shellac resin is then obtained by hot filtration. The shellac resin can be produced either as membrane, flakes, buttons, die, wax, or refined products including waxed, dewaxed, bleached and de-colored liquids [6-9].

Crude shellac is constituted from resin, wax and dye [10] contaminated by wood particles, sand, insect bodies, etc. It was reported by Sharma [4] that the crude shellac contains about 68 % resin, 6% wax, 1 % dye, and about 25 % contaminants.

Chemically, the shellac resin is consisted of several polar and non-polar constituents. This resin can be divided into two component, namely soft resin (25% wt/wt) constituted mainly from ether-soluble monoesters, and hard resin (75% wt/wt) which is ether-insoluble complex [11, 12]. Furthermore, the dyestuffs in the shellac are classified into shellac dye (Laccaic acid) that is soluble in alkali solution and erythrolaccin which is soluble in alcohol [13]. In addition, shellac matrix is a polymeric complex constituted from mono- and polyesters of hydroxy aliphatic and sesquiterpenoid acids [12]. It was indicated that the main ester components of shellac are aleuritic acid, jalaric, laccijalaric acids, and butolic acid [5, 14-18]. In addition, chemical composition of shellac seems nearly to be constant, although some components [19, 20] changes quantitatively according to nature of their host trees………………………………….

The modified introduction:

Shellac is a polymeric resin secreted by the female lac insect (Laccifer lacca), which can be found on certain trees, especially Schleichera oleosa, Schleichera oleos, Ziziphus mauritania and Butea monosperma. The immature nymphs exude a resin/wax mixture through their bodies after they suck sap from the tree, forming a defensive cell of a cocoon [1]. Shellac is processed and sold as dry flakes and still occupies an important position among natural resins [2–4].

Shellac manufacturing involves the following processes: (a) tree pruning, (b) inoculation to introduce lac insects onto suitable host plants, (c) the removal of phunki lac, (d) harvesting, and (e) the scraping of crude flakes (CFs) (seedlac) along with impurities such as sand, dirt, bark and wood particles, and insect bodies [4,5]. The ordinary seedlac purification process comprises washing, hot filtration and drying. Shellac resin can be produced either as membranes, flakes, buttons, dyes, wax or refined products, including waxed, dewaxed and bleached liquids [6–9].

The CFs are composed of resin (approximately 68%), wax (approximately 6%), around 1% dye and around 25% contaminants [4,10]. Chemically, shellac resin comprises soft resin (ether-soluble monoesters) and hard resin (ether-insoluble complex) in a ratio of approximately 1:3 [11,12]. ….. [30].

For dentistry applications, shellac is used for partial dentures and frameworks, clasps, primary crowns and bridges, full dentures, orthodontic appliances, anti-snoring devices and various types of mouth guards and splints [31]. It has been reported that shellac base plates are useful in constructing special trays and temporary denture bases [32,33]. In addition, bleached shellac and shellac wax are essential natural polymers in tooth varnish composites. Existing tooth varnish compositions generally contain an active component to treat caries, provide fluoride therapy, treat xerostomia and tooth sensitivity, and/or whiten or bleach teeth, as well as an adhesive film-forming component to cause the active material to adhere to the tooth. Bleached shellac and shellac wax can be included in the varnish formula from approximately 0.1–20% and approximately 25% of the weight of the composition, respectively [34]. Furthermore, permanent tooth varnishing is especially recommended for patients who are struggling with problems such as tooth sensitivity, xerostomia (dry mouth syndrome), orthodontic treatment or an inability to maintain proper oral hygiene, such as in patients with mental illness [35].

One disadvantage of tooth varnishes is their tendency to be multiphase, whereby the active component is insoluble in the adhesive film-forming phase, and the varnish may be separated out into distinct phases. Additionally, components of the adhesive film-forming phase may also separate into distinct phases over time. Users typically need to stir the varnish in order to mix the phases, which is time consuming and wasteful, as the varnish adheres to the mixing apparatus and is then discarded. Thus, there exists a need to develop tooth varnish compositions with greater stability, wherein the phases do not readily separate [34].

The addition of fluoride ion sources (at least approximately 5000 ppm) is essential for tooth varnish using soluble salts of fluoride ions; for example, sodium fluoride, potassium fluoride, calcium fluoride and zinc fluoride. An antibacterial agent may be included in the tooth varnish formula using benzoic acid or one of its salts. Furthermore, an anti-sensitivity agent may be incorporated using either potassium, zinc or chloride salts, as well as capsaicin or eugenol. Such agents may be added in effective amounts (1–20% wt/wt).

The varnish may contain a tooth whitener (such as hydrogen peroxide) and a suitable non-toxic solvent, such as ethanol. This solvent may also function as a viscosity modifier, and to ensure an even deposition of the film-forming component [34].

A shellac varnish formula was created by Hoang-Dao et al. [35] that showed adequate cellular compatibility and a significant effect on human dentin hydraulic conductance. This indicates that the new material is safe and seems to be effective as a potential desensitizing agent.

The aims of the present work were to invent a procedure and an apparatus using microwave irradiation beams for the purification of shellac CFs into a purer and more reliable raw material suitable for dental varnish.

Point 7: He consider that the cited references relevant to the research must be improved.

Response 7: The cited references relevant to the research were improved by adding the following references.

  • Simunkova, K.; Pánek, M.; Zeidler, A. Comparison of selected properties of shellac varnish for restora-tion and polyurethane varnish for reconstruction of historical artefacts. Coatings 2018, 8, 119.
  • The determination of moisture in shellac. Ind. Eng. Chem. 1915, 7, 633.
  • K.; Wen, F.H.; Yu, D.G.; Yang, Y.; Zhang, D.F. Electrosprayed hydrophilic nanocomposites coated with shellac for colon-specific delayed drug delivery. Mater. Des. 2018, 143, 248–255.
  • Dental Varnishing. Medicover. 2022. www.medicover.pl/en/dentistry/dental-varnishing/
  • Cahyanto, A.; Marwa, D.F.; Saragih, K.; Takarini, V.; Hasratiningsih, Z. Enamel remineralization effect using dewaxed shellac varnishes with added carbonate apatite and tricalcium phosphate. J. Int. Dent. Med. Res. 2020, 13, 533–538.

In addition, numbers of citations were moved through a new rearrangement within the article.

Point 8:  He consider that the research design must be improved.

Response 8: The research design was improved sufficiently in the previous correction round by doing the following actions: 

  1. The title of “3. Illustration of the Microwave Behavior and Effect”was considered as an extent to to the discussion section not as a separate sub-subtitle.
  2. Future Perspectives was included into the conclusions section to highlight the emergency needs to improve the patent benefits and reduces any limitations in its applications as future targets.

Point 9: He consider that the conclusions supported by the results must be improved.

Response 9:

Old “conclusions”

Crude shellac, thermoplastic polymer, was subjected to a novel refining process termed as microwaved-thermospinning to obtain more purified- and homogeneous-end product free of physical contaminants which has revealed to synthesis of a novel product, namely shellac floss. Two different thermo-spinners, heated by either electric heating coils or microwave irradiation individually were used to convert shellac flakes into floss. To confirm that both thermospinning techniques did not distort the parent quality of shellac, the three shellac samples (crude flakes and electric- and micro-waved-thermospun flosses) were characterized by determining their chemical and physical properties including x-ray diffraction as well as Fourier-transform infrared spectroscopy.

All the mean values of the properties studied for the three types of shellac were found to be in the standard ranges. Although there are no statistical differences between the three shellac flosses types in their properties except for the IHA and AV, the micro-waved-thermospun floss exhibited the best type among the other types for all the properties studied. For the physical properties of shellac, their mean values ranges were about 13.2 %-13.6 %, 1.181-1.183, 52.01-53.3 mm, and 33.7 min.-33.81 min. for color index, specific gravity, flow of shellac molten, and heat of polymerization, respectively. The X-ray diffractograms of the shellac types, namely crude flakes, elec-tric-thermospun floss, microwaved-thermospun flosses, there are three essential peaks were detected at two theta of 11⁰ (broad peak), 17.5 ⁰ (broad peak), and 22 ⁰ (relatively sharp), respectively indicating the amorphicity of shellac.

The mean values ranges of the chemical properties of shellac were found to be about 1.1%-1.95 %, 0.231 %-0.287 %, 2.3 % to 2.71 %, and (68.11-69.79 mg KOH g for insolubility in hot alcohol, ash content, waxiness content, and acid value, respectively. The largest FTIR bands were recorded at 1252, 1713, 2856, 2933, and 2995 cm−1 for all the shellac types, namely crude flakes, electric- and microwave-thermospun flosses. Accordingly, since there is no change of molecular structure of shellac observed by FTIR spectroscopy along the three samples type, thermospinning process, else microwaving or electro-heating did not affect the chemical composition of the parent shellac.

Microwaved-thermospinner did not distort the parent shellac quality and gave shellac floss with relatively higher quality compared to that obtained by electro-thermospinner. It allows to construct larger and cheaper machines that permit mass production of shellac floss that endorses the suitability of the invention such applications.

In the near future, floss may become the most handled type of shellac instead of flakes and other types of shellac in the market due to its high purity and ease and low cost of manufacturing and transportation. Importantly, morphological, mechanical and cytocompatibility characterization techniques must be studied and employed to show the applicability of those materials. Encapsulation of nanoforms of drugs in the shellac fibers using improvisation of electrospinning techniques must be extended to the microwaved-thermospun floss. Determination of the best formulations of shellac compo-sites and their mechanical properties is a cornerstone in the dental and pharmaceutical fields.

Modified “Conclusions and Future Perspectives

Crude shellac, a thermoplastic polymer, was subjected to a novel refining process, termed microwave thermospinning, to obtain a more purified and homogeneous end product free of physical contaminants (shellac floss). Two different thermospinners, heated by either electric heating coils or microwave irradiation, were individually used to convert shellac flakes into floss. To confirm that both thermospinning techniques did not degrade the parent quality of shellac, three shellac types (crude flakes, electro-thermospun floss and microwave-thermospun floss) were characterized physically and chemically.

The mean values of the properties studied for the three types of shellac were all found to be in the standard ranges. Although there were no statistical differences among the three shellac floss types regarding their properties, except for insolubility in hot alcohol, acid value and moisture content, the microwave-thermospun floss exhibited a superior material for almost all the properties studied. For the physical properties of shellac, the mean value ranges were approximately 13.2–13.6%, 1.181–1.183, 52.01–53.3 mm and 33.7–33.81 min for color index, specific gravity, flow of molten shellac and heat of polymerization, respectively. Each X-ray diffractogram of the three shellac types had three essential peaks detected at two theta of 11° (broad peak), 17.5° (broad peak), and 22° (relatively sharp) for flakes, and electric and microwave thermospun flosses, respectively, indicating the amorphicity of the shellac.

The mean value ranges of the chemical properties of shellac were found to be approximately 1.1–1.95%, 0.231–0.287%, 2.3–2.71% and 68.11–69.79 mg KOH/g for in-solubility in hot alcohol, ash content, waxiness content and acid value, respectively. The largest FTIR bands were recorded at 1252, 1713, 2856, 2933 and 2995 cm−1 for the three shellac types. Accordingly, the thermospinning process did not affect the molecular structure of the parent shellac.

The microwave thermospinner did not distort the parent shellac quality and provided a relatively higher quality shellac floss compared with that obtained using the electric thermospinner. This result allows for the construction of larger and cheaper machines that permit the mass production of shellac floss, endorsing the suitability of the invention of such applications.

On the other hand, the microwave beam must be transmitted within well isolated waveguides to protect labors from radiation.

In conclusion, it is expected that using bleached microwaved shellac floss along with shellac wax in proper amounts can improve varnish stability at higher temperatures. In addition, the formulated tooth varnish is expected to possess better adhesive ability, be more easily applied to teeth, result in less coloring of teeth surfaces and retain the activity of the active component (fluoride ions) compared with those tooth varnishes that contain ordinary shellac flakes.

In the near future, floss may become the most handled type of shellac, instead of flakes or other types of shellac in the market, due to its high purity and ease of use, and its low cost of manufacturing and transportation. Importantly, morphological, mechanical and cytocompatibility characterization techniques must be studied and employed to show the applicability of these materials. The encapsulation of nanoforms of drugs in shellac fibers using improvised electrospinning techniques must be extended to microwave-thermospun floss. Determining the best formulations of shellac compo-sites and their mechanical properties is essential in the dental and pharmaceutical fields. In addition, there is a need to develop dental varnish compositions that do not adversely affect the color of teeth and that increase tooth stability.

In addition, it has also several agricultural uses, especially for seeds coating and leather finishes.

Reviewer 2 Report

Authors have considered this reviewer feedback and made effort to improve the manuscript.  Many of the comments previously made have been addressed.  However, the  introduction remains overlong, still having much detail of how shellac is produced and what it is constituted of.  The analytical  details  provided for the  input material to the floss creation is of value but might be edited with all but the absolutely key data being  consigned to a a supplementary information section.  The potential uses of the floss in dental and pharmaceutical applications have been elaborated just a little based on this reviewer's recommendation that this is a need.  However, I would still like to see that developed just a little further to say exactly how the material might be applied in dental and pharmaceutical fields, e.g. a drug carrier to treat periodontal disease, a way of providing a drug/polymer composite  carrying  possibly amorphous drug to deliver orally in a manner dependent on the solubility properties of shellac in the gastrointestinal tract.

Authors are asked to consider further these broad  manuscript improvement suggestions in  thinking about its revision for submission. 

Author Response

(The authors gave the same response as above.)

Round 3

Reviewer 1 Report

I consider the article does not meet the quality and novelty to be published in Polymers. I recommend the submission to a lower impact factor journal.

Author Response

Based on the comments of the reviewer 1 that the introduction, the research design, the methods, the results and the conclusions must be improved, our response is as follow:

For work target:

This aim was modified to be as follow:

  1. To invent procedure and apparatus using microwave irradiation beams for the purification of shellac CFs into a purer and more reliable raw material suitable for different applications.
  2. Bleaching the MTS-shellac floss using sodium hypochlorite.
  3. Exploring suitable dental applications of the obtained bleached MTS-shellac floss based on its quality.

For the research design

New paragraphs under a separate title were added concerning to the bleached shellac required to fabricate colorless/white dental products for each of the five principle sections of the article (the introduction, the materials and methods, the results, the discussion, the conclusions).

For the introduction section:

It was compressed, idea-arranged, some references were added, especially for the added-bleached shellac title.

For the Materials and Methods:

New titles were added belonging to bleached shellac, namely

“2.5. Bleaching the MTS-Shellac Floss”.

“2.5.1. Preparation of the Bleached Shellac (BS) films”.

“2.5.2. Characterization of the BS-Film”

Several individual paragraphs were included in this sub-section (without numbering) to cover the characterization procedures of the bellow-mentioned properties of the BS.

Six effective properties of the BS were investigated, namely color change (CC), acid values (AV), insoluble solid matter (ISM), moisture content (MC), ash content (AC) and Young’s modulus.

For the results section:

New titles were added belonging to bleached shellac, namely

“3.3. Properties of the BS-Film”. Four paragraphs comprising the four properties of the bleached shellac studied (without numbering) beside a fifth paragraph concerning to a general comment for the obtained bleached shellac film.

For the presentation of figures:

  • Figure 6 was deleted.
  • As shown in Figure 1 presented above, the red circled-microwave shape was extracted from the deleted-figure 6 and inserted into it.

For the presentation of table:

New table was added presenting as follow:

Table 3. Mean values 1,2 of the important properties of the unbleached and the bleached shellac (BS) film produced from microwave thermospun (MTS) shellac floss (SF), namely color change (CC), acid value (AC), insoluble solid matter (ISM) and mechanical properties.

Property

Unbleached parent

MTS-SF

BS-film

CC, %

0

95.74 [1.27]

AV, mg KOH/g

68.41 [1.52]

72.01 [2.47]

ISM, %

MC, %

AC, %

4.58 [0.46]

1.31 [0.04]

0.287 [0.061]

1.82 [0.22]

1.03 [0.08]

0.216 [0.083]

Young’s modulus, MPa

10.14 [0.61]

9.88 [0.61]

1 Each value is an average of 5 samples, 2 Values within parentheses are standard deviations.

For the Discussion section:

  • An individual main title was added to this section, namely “Properties of the BS-Film” that discuss the characteristic results of the bleached shellac (BS). There are four paragraphs representing the four properties studied, namely color change (CC), acid values (AV), insoluble solid matter (ISM), moisture content (MC), ash content (AC) and Young’s modulus (E).
  • Another individual sub-title was added to this section, namely “4.4. Suitability of the Resulted Shellac Forms for Some Dental Applications” that discuss the application of the produced shellac film in some important dental products such as tooth varnish, baseplates, ….
  • The title of “Illustration of the Microwave Behavior and Effect” that found at the end of the discussion was deleted as a title, but its text content was concluded, compressed and moved to begin the discussion section.

For the conclusions section:

  • Some statements were merged together to contract the volume of this section.
  • One paragraph concerning to the bleached shellac was added to this section.

For the cited references

One reference was deleted, while twelve references were added to this article to cover importance of the bleached shellac in some dental applications.

References deleted:

Hindi, S.S.Z. Some crystallographic properties of cellulose I as affected by cellulosic resource, smoothing, and computation methods. Int. J. Innov. Res. Sci. Eng. Technol. 2017, 6, 732–752.

References added

  1. Soradech, S., Nunthanid, J., Limmatvapirat, S., & Luangtana-anan, M. Utilization of shellac and gelatin composite film for coating to extend the shelf life of banana. Journal of Food Control Part B, 2017, 73, 1310–1317.
  2. Klineberg, I and Earnshawt, R. Physical properties of shellac baseplate materials. 468 Australian Dental Journal, October, I967.
  3. Cockeram, H.S., Levine, S.A. The Physical and Chemical Properties of Shellac. J. Soc. Cosmet. Chem. 12: 316-323 (1961).
  4. Trezza, T.A., Krochta, J.M. Specular Reflection, Gloss, Roughness and Suface Heterogeneity of Biopolymer Coatings. J. Appl. Polym. Sci. 79: 2221-2229 (2001).
  5. Penning, M. (1996). Aqueous shellac solutions for controlled release coatings: Chemical aspects of drug delivery systems, The Royal Society of Chemistry. 178, 146–154.
  6. Abou-Okeil, A., El-Shafie, A., & El Zawahry, M. M. (2010). Ecofriendly laccase–hydrogen peroxide/ultrasound-assisted bleaching of linen fabrics and its influence on dyeing efficiency. Ultrasonics Sonochemistry, 17, 383–390.
  7. Xu, C., Hinks, D., Sun, C., & Wei, Q. (2015). Establishment of an activated peroxide system for low-temperature cotton bleaching using N-[4 (triethylammoniomethyl)benzoyl] butyrolactam chloride. Carbohydrate Polymers, 119, 71–77.
  8. Osman, Z. (2012). Investigation of different shellac grades and improvement of release from air suspension coated pellets. Dissertation, Department of Chemistry, Gutenberg-Universität, Mainz.
  9. Abdel-Halim, E. S. (2012a). An effective redox system for bleaching cotton cellulose. Carbohydrate Polymers, 90, 316–321.
  • Abdel-Halim, E. S. (2012b). Simple and economic bleaching process for cotton fabric. Carbohydrate Polymers, 88, 1233–1238.
  • Abdel-Halim, E. S., & Al-Deyab, S. S. (2013). One-step bleaching process for cotton fabrics using activated hydrogen peroxide. Carbohydrate Polymers, 92, 1844–1849.
  • Ibrahim, N. A., Sharaf, S. S., & Hashem, M. M. (2010). A novel approach for low temperature bleaching and carbamoylethylation of cotton cellulose. Carbohydrate Polymers, 82, 1248–1255.

Reviewer 2 Report

The authors have still not connected the work described in the manuscript with the title. The work described is all around preparing and characterizing the spun shellac fibres, and in fact this is now done  in a satisfactory manner. There is no work describing how the floss is applied as dental varnish.  There is  speculation as to this possible use, but that is all. The authors should either take out the reference to dental varnish in the title, and still address how to indicate this  potential use, maybe among others, in the text, or keep the title as is and add a brief section showing how fluoride,  chlorhexidine or peroxide can be incorporated into the floss.

Author Response

Comments and Suggestions of the Reviewer 1:

Based on the comments of the reviewer 1 that the introduction, the research design, the methods, the results and the conclusions must be improved, our response is as follow:

For work target:

This aim was modified to be as follow:

  1. To invent procedure and apparatus using microwave irradiation beams for the purification of shellac CFs into a purer and more reliable raw material suitable for different applications.
  2. Bleaching the MTS-shellac floss using sodium hypochlorite.
  3. Exploring suitable dental applications of the obtained bleached MTS-shellac floss based on its quality.

For the research design

New paragraphs under a separate title were added concerning to the bleached shellac required to fabricate colorless/white dental products for each of the five principle sections of the article (the introduction, the materials and methods, the results, the discussion, the conclusions).

For the introduction section:

It was compressed, idea-arranged, some references were added, especially for the added-bleached shellac title.

For the Materials and Methods:

New titles were added belonging to bleached shellac, namely

“2.5. Bleaching the MTS-Shellac Floss”.

“2.5.1. Preparation of the Bleached Shellac (BS) films”.

“2.5.2. Characterization of the BS-Film”

Several individual paragraphs were included in this sub-section (without numbering) to cover the characterization procedures of the bellow-mentioned properties of the BS.

Six effective properties of the BS were investigated, namely color change (CC), acid values (AV), insoluble solid matter (ISM), moisture content (MC), ash content (AC) and Young’s modulus.

For the results section:

New titles were added belonging to bleached shellac, namely

“3.3. Properties of the BS-Film”. Four paragraphs comprising the four properties of the bleached shellac studied (without numbering) beside a fifth paragraph concerning to a general comment for the obtained bleached shellac film.

For the presentation of figures:

  • Figure 6 was deleted.
  • As shown in Figure 1 presented above, the red circled-microwave shape was extracted from the deleted-figure 6 and inserted into it.

For the presentation of table:

New table was added presenting as follow:

Table 3. Mean values 1,2 of the important properties of the unbleached and the bleached shellac (BS) film produced from microwave thermospun (MTS) shellac floss (SF), namely color change (CC), acid value (AC), insoluble solid matter (ISM) and mechanical properties.

Property

Unbleached parent

MTS-SF

BS-film

CC, %

0

95.74 [1.27]

AV, mg KOH/g

68.41 [1.52]

72.01 [2.47]

ISM, %

MC, %

AC, %

4.58 [0.46]

1.31 [0.04]

0.287 [0.061]

1.82 [0.22]

1.03 [0.08]

0.216 [0.083]

Young’s modulus, MPa

10.14 [0.61]

9.88 [0.61]

1 Each value is an average of 5 samples, 2 Values within parentheses are standard deviations.

For the Discussion section:

  • An individual main title was added to this section, namely “Properties of the BS-Film” that discuss the characteristic results of the bleached shellac (BS). There are four paragraphs representing the four properties studied, namely color change (CC), acid values (AV), insoluble solid matter (ISM), moisture content (MC), ash content (AC) and Young’s modulus (E).
  • Another individual sub-title was added to this section, namely “4.4. Suitability of the Resulted Shellac Forms for Some Dental Applications” that discuss the application of the produced shellac film in some important dental products such as tooth varnish, baseplates, ….
  • The title of “Illustration of the Microwave Behavior and Effect” that found at the end of the discussion was deleted as a title, but its text content was concluded, compressed and moved to begin the discussion section.

For the conclusions section:

  • Some statements were merged together to contract the volume of this section.
  • One paragraph concerning to the bleached shellac was added to this section.

For the cited references

One reference was deleted, while twelve references were added to this article to cover importance of the bleached shellac in some dental applications.

References deleted:

Hindi, S.S.Z. Some crystallographic properties of cellulose I as affected by cellulosic resource, smoothing, and computation methods. Int. J. Innov. Res. Sci. Eng. Technol. 2017, 6, 732–752.

References added

  1. Soradech, S., Nunthanid, J., Limmatvapirat, S., & Luangtana-anan, M. Utilization of shellac and gelatin composite film for coating to extend the shelf life of banana. Journal of Food Control Part B, 2017, 73, 1310–1317.
  2. Klineberg, I and Earnshawt, R. Physical properties of shellac baseplate materials. 468 Australian Dental Journal, October, I967.
  3. Cockeram, H.S., Levine, S.A. The Physical and Chemical Properties of Shellac. J. Soc. Cosmet. Chem. 12: 316-323 (1961).
  4. Trezza, T.A., Krochta, J.M. Specular Reflection, Gloss, Roughness and Suface Heterogeneity of Biopolymer Coatings. J. Appl. Polym. Sci. 79: 2221-2229 (2001).
  5. Penning, M. (1996). Aqueous shellac solutions for controlled release coatings: Chemical aspects of drug delivery systems, The Royal Society of Chemistry. 178, 146–154.
  6. Abou-Okeil, A., El-Shafie, A., & El Zawahry, M. M. (2010). Ecofriendly laccase–hydrogen peroxide/ultrasound-assisted bleaching of linen fabrics and its influence on dyeing efficiency. Ultrasonics Sonochemistry, 17, 383–390.
  7. Xu, C., Hinks, D., Sun, C., & Wei, Q. (2015). Establishment of an activated peroxide system for low-temperature cotton bleaching using N-[4 (triethylammoniomethyl)benzoyl] butyrolactam chloride. Carbohydrate Polymers, 119, 71–77.
  8. Osman, Z. (2012). Investigation of different shellac grades and improvement of release from air suspension coated pellets. Dissertation, Department of Chemistry, Gutenberg-Universität, Mainz.
  9. Abdel-Halim, E. S. (2012a). An effective redox system for bleaching cotton cellulose. Carbohydrate Polymers, 90, 316–321.
  • Abdel-Halim, E. S. (2012b). Simple and economic bleaching process for cotton fabric. Carbohydrate Polymers, 88, 1233–1238.
  • Abdel-Halim, E. S., & Al-Deyab, S. S. (2013). One-step bleaching process for cotton fabrics using activated hydrogen peroxide. Carbohydrate Polymers, 92, 1844–1849.
  • Ibrahim, N. A., Sharaf, S. S., & Hashem, M. M. (2010). A novel approach for low temperature bleaching and carbamoylethylation of cotton cellulose. Carbohydrate Polymers, 82, 1248–1255.

Comments and Suggestions of the Reviewer 2:

The authors have still not connected the work described in the manuscript with the title. The work described is all around preparing and characterizing the spun shellac fibers, and in fact this is now done in a satisfactory manner. There is no work describing how the floss is applied as dental varnish.  There is speculation as to this possible use, but that is all. The authors should either take out the reference to dental varnish in the title, or still address how to indicate this potential use, maybe among others, in the text, or keep the title as is and add a brief section showing how fluoride, chlorhexidine or peroxide can be incorporated into the floss.

The authors appreciate the recognition of the reviewer 2 in which our work described is now done in a satisfactory manner. In addition, the reviewer 2’s comments are appreciated, whereby all the cited references, the research design, the methods and the results are appropriate.

The title was modified to be “A Novel Microwave-Thermospun Shellac Floss: Synthesis and Bleaching for Some Dental Applications”.

Based on the comments of the reviewer 2 that the introduction and the conclusions can be improved, our response is presented above in my response to the reviewer 1’s comments.

The 1st aim of the following targets of the present investigation was added, while the 2nd aim was modified to cover some dental applications of the obtained bleached MTS-shellac floss based on its quality.

  1. Bleaching the MTS-shellac floss using sodium hypochlorite.
  2. Exploring suitable dental applications of the obtained bleached MTS-shellac floss based on its quality.
